# Cell-type specific, inducible and acute degradation of targeted protein in mice by two degron systems

Motoi Yamashita [1], Chihiro Ogawa[1], Baihao Zhang [2], Tetsuro Kobayashi[3], Aneela Nomura [1], Clive Barker[4], Chengcheng Zou[1], Satoshi Yamanaka [5,6], Ken-ichiro Hayashi[7], Yoichi Shinkai [8], Kazuyo Moro[3], Sidonia Fargarasan [2], Koshi Imami [4,9], Jun Seita [4], Fumiyuki Shirai [10], Tatsuya Sawasaki [5], Masato T. Kanemaki [11,12,13] & Ichiro Taniuchi [1] ✉

Despite its broad application in in vitro studies, the application of targeted protein degradation (TPD) to animal models faces considerable challenges. Here, we develop inducible and cell-type specific TPD systems in mice using two degron systems: *Oryza sativa* TIR1$^{F74G}$ (OsTIR1)-auxin-inducible degron 2 (AID2) and human cereblon (hCRBN)-SALL4 degron (S4D). Efficient degradation of Satb1$^{Venus}$ protein by these systems recapitulates phenotypes observed in the Satb1-deficient mice. These TPD are successfully applied in both the fetal and neonatal stages. The OsTIR1-AID2 system proves to be effective for membrane proteins such as PD-1, emulating the effects of the anti-PD-1 antibody. Degradation of Bcl11b reveals a role of Bcl11b which was not characterized by the Cre-loxP system. Collectively, in vivo TPD technologies developed in this study enable inducible, temporal, and cell type-specific depletion of target proteins with high efficacy in mice. These technologies have a wide range of applications in the diverse fields of biological and medical research.

Although genome engineering technologies, such as Cre-loxP mediated gene inactivation, are powerful tools for investigating the consequences of the loss of protein-of-interest (POI) expression in in vivo models, they have intrinsic limitations such as inability of recovery of protein expression and lagged response for loss of protein after gene inactivation, and their dependence on availability for appropriate Cre transgenic line. Thus, the development of a method for temporary loss of POI in animal models with high efficacy, well-characterized kinetics and cell-type specificity promises significant advantages for diverse biological and medical studies.

Currently, several methods have emerged as effective tools for targeted protein degradation (TPD) by utilizing proteasome-mediated degradation, and have been extensively used to investigate protein function in vitro[1,2]. One prominent example is the cereblon (CRBN) E3

[1]Laboratory for Transcriptional Regulation, RIKEN Center for Integrative Medical Sciences (IMS), Yokohama, Kanagawa, Japan. [2]Laboratory for Mucosal Immunity, RIKEN Center for Integrative Medical Sciences (IMS), Yokohama, Kanagawa, Japan. [3]Laboratory for Innate Immune Systems, RIKEN Center for Integrative Medical Sciences (IMS), Yokohama, Kanagawa, Japan. [4]Laboratory for Integrative Genomics, RIKEN Center for Integrative Medical Sciences (IMS), Yokohama, Kanagawa, Japan. [5]Division of Cell-Free Sciences, Proteo-Science Center, Ehime University, Matsuyama, Ehime, Japan. [6]Division of Proteo-Interactome, Proteo-Science Center, Ehime University, Matsuyama, Ehime, Japan. [7]Department of Biochemistry, Okayama University of Science, Okayama, Okayama, Japan. [8]Cellular Memory Laboratory, RIKEN Cluster for Pioneering Research, Wako, Saitama, Japan. [9]Proteome Homeostasis Research Unit, RIKEN Center for Integrative Medical Sciences (IMS), Yokohama, Kanagawa, Japan. [10]Drug Discovery Chemistry Platform Unit, RIKEN Center for Sustainable Resource Science, Wako, Saitama, Japan. [11]Department of Chromosome Science, National Institute of Genetics, Mishima, Japan. [12]Graduate Institute for Advanced Studies, SOKENDAI, Mishima, Japan. [13]Department of Biological Science, The University of Tokyo, Bunkyo-ku, Japan. ✉e-mail: ichiro.taniuchi@riken.jp

ubiquitin ligase complex (CRL4$^{CRBN}$)[3]. Thalidomide and its derivatives, such as pomalidomide (POM), serve as a molecular glue, facilitating the binding of non-natural substrates to CRL4$^{CRBN}$, known as neo-substrates, thereby altering the substrate specificity of the CRL4$^{CRBN}$ ubiquitin ligase[4]. In a recent study, a human CRBN (hCRBN)-binding domain within Sal-like protein 4 (SALL4) was identified and termed as the SALL4 degron (S4D)[5], which comprises 29 amino acids derived from the second zinc finger of SALL4. In the presence of thalidomide or its derivatives, POI tagged with S4D is bound by hCRBN, followed by the ubiquitination and sequential degradation of POI.

Another TPD system utilizes a plant derived E3 ubiquitin ligase complex. F-box transport inhibitor response 1 (TIR1) is a plant protein with E3 ubiquitin ligase activity that naturally recognizes the plant hormone, auxins, also known as indole-3-acetic acid (IAA). Consequently, the introduction of exogenously expressed *Oryza sativa* TIR1 (OsTIR1) into non-plant cells has been demonstrated to effectively degrade POI when tagged with an *Arabidopsis thaliana* IAA17-derived degron with the supplementation of IAA[6]. This degron tag, comprising 70 amino acids, is referred to as the mini-auxin-inducible degron (mAID). The AID system was modified to decrease a basal degradation in the absence of IAA and to augment ubiquitylation activity in the presence of IAA, thus generating a second version of AID (AID2) using OsTIR1$^{F74G}$ and 5-phenyl-indole-3-acetic acid (5-Ph-IAA)[7].

Despite the initial adoption of degron technologies in in vivo mouse models, the effectiveness and utility of degron systems in vivo remain unexplored[8–12]. In this study, we generate two transgenic mouse lines that express *hCRBN* or *OsTIR1$^{F74G}$* transgene from the *Rosa26* locus upon removal of the loxP-Stop-loxP (LSL) sequences by expression of Cre recombinase, thereby enabling cell type-specific expression of these E3 ubiquitin ligases. We demonstrate the rapid, inducible, and temporal degradation of nuclear proteins. Notably, this degradation is observed in the brain, as well as in embryos and neonates after treatment with mothers, and is applied to plasma membrane proteins.

## Results

### hCRBN-S4D and OsTIR1-AID2 systems in mice

To assess the capacity of the hCRBN-S4D and OsTIR1-AID2 systems to degrade POI in mice, we chose a nuclear protein, Special AT-rich binding protein 1 (Satb1). This choice was influenced by our development of a mouse strain possessing a *Satb1$^{Venus}$* allele, which produces a Satb1$^{Venus}$ fusion protein, enabling easy monitoring of its expression by flow cytometry[13]. Utilizing CRISPR/Cas9 genome-editing technology, we introduced S4D (87 bp, 29 aa) or mAID (210 bp, 70 aa) sequences between the Venus and *Satb1* coding sequence of *Satb1$^{Venus}$* allele. This approach successfully generated *Satb1$^{Venus-S4D}$* and *Satb1$^{Venus-mAID}$* alleles, hereafter referred to as *Satb1$^{V-S4D}$* and *Satb1$^{V-AID}$* alleles (Supplementary Fig. 1a). Immunoblot and flow cytometric analyses confirmed the expression of Satb1$^{V-S4D}$ and Satb1$^{V-AID}$ fusion proteins from the respective alleles (Supplementary Fig. 1b, c). However, a decrease in Venus fluorescence intensity was observed, with the mean fluorescence intensity (MFI) at 67.7% and 67.6% in CD4$^+$CD8$^+$ double-positive (DP) thymocytes of *Satb1$^{V-S4D/V-S4D}$* and *Satb1$^{V-AID/V-AID}$* mice, respectively, compared to *Satb1$^{Venus/Venus}$* controls. In *Satb1$^{V-AID/V-AID}$* mice, there was a lower fraction of CD4 single-positive (SP) mature thymocytes and higher fraction of CD8 SP mature thymocytes (Supplementary Fig. 1d, e). Importantly, Satb1 function was not apparently affected by the insertion of either degron tag, since the two known phenotypes of loss of Satb1, emergence of aberrant mature DP thymocytes and *Foxp3* de-repression observed in *Satb1$^{Flox/Flox}$;Cd4Cre* mice[14,15], were not observed in either *Satb1$^{V-S4D/V-S4D}$* or *Satb1$^{V-AID/V-AID}$* mice (Supplementary Fig. 1d). We then established two transgenic mouse lines, *Rosa26$^{LSL-hCRBN}$* and *Rosa26$^{LSL-OsTIR1}$*, which expressed *hCRBN* and *OsTIR1$^{F74G}$* from the *Rosa26* locus upon excision of the LSL sequence via Cre recombinase-mediated site-specific recombination (Fig. 1a and Supplementary

Fig. 1f). An ires-EGFP sequence following the coding sequences for *OsTIR1$^{F74G}$* or *hCRBN* served as a maker for the identification of cells expressing OsTIR1$^{F74G}$ or hCRBN, which manifested as EGFP$^+$ cells. These *Rosa26$^{LSL-hCRBN}$* and *Rosa26$^{LSL-OsTIR1}$* mice were crossed with *Cd4Cre* and *VavCre* mice to generate mouse strains that selectively expressed these transgenes in the T cell lineage and all hematopoietic cell lineages, respectively. Additionally, these strains were crossed with *EIIaCre* strain to generate mouse lines that systemically expresses *hCRBN* and *OsTIR1$^{F74G}$* transgenes which are hereafter referred to as *Rosa26$^{hCRBN}$* and *Rosa26$^{OsTIR1}$*, respectively (Supplementary Fig. 1g). Consistent with previous reports, OsTIR1$^{F74G}$ exhibited negligible basal degradation, as indicated by the comparable *Satb1$^{V-AID}$* levels in the presence or absence *of OsTIR1$^{F74G}$* (Supplementary Fig. 1h).

Sixteen hours after intraperitoneal administration of 5-Ph-IAA or POM in *Rosa26$^{OsTIR1/+}$;Satb1$^{V-AID/V-AID}$* or *Rosa26$^{hCRBN/+}$;Satb1$^{V-S4D/V-S4D}$* mice, western blot analysis indicated efficient reduction of Satb1$^{V-AID}$ and Satb1$^{V-S4D}$ protein levels in DP thymocytes (Fig. 1b) and flow cytometry estimated that Venus MFI was reduced to approximately 9.1% and 6.0% of their untreated levels, respectively (Fig. 1c, d).

Next, we investigated the dynamic kinetics of Satb1 protein loss and recovery in circulating T cells in the peripheral blood of individual *Rosa26$^{OsTIR1/+}$;Satb1$^{V-AID/V-AID}$* mice following a single intraperitoneal administration of 5-Ph-IAA. The intensity of Venus started to decline 2 h post-injection and further declined in a biphasic pattern from 3 to 6 h (Fig. 1e, f). The reduction in Venus intensity reached a maxima level at 8 h post-administration. Recovery of Satb1$^{V-AID}$ levels was observed in 28–32 h, necessitating 72 h to fully restore to the baseline levels. We similarly elucidated the dynamic kinetics of Satb1$^{V-S4D}$ degradation in *Rosa26$^{hCRBN/+}$;Satb1$^{V-S4D/V-S4D}$* mice after an intraperitoneal injection of POM in PBS containing 10% dimethyl sulfoxide (DMSO). Compared to the OsTIR1-AID2 system, Venus expression levels remained low for a prolonged period, spanning 10–13 days after the POM injection (Fig. 1e, f). Undissolved POM formed aggregates in the peritoneal cavity following intraperitoneal injection, resulting in a prolonged POM supply that would cause sustained Satb1 degradation (Supplementary Fig. 2a). Thus, we changed the solvent to 15% DMSO, 17.5% Cremophor EL, 8.75% Ethanol, 8.75% HCO-40, and 50% PBS[16] to improve the solubility of POM (Supplementary Fig. 2a). Administration of POM in this alternative solvent resulted in the rapid degradation of Satb1$^{V-S4D}$, reaching a maximum reduction in 2 h, with a quicker recovery to baseline at 24 h post-administration (Fig. 1e, f). The hCRBN-S4D system exhibited sharper kinetics characterized by rapid degradation and swift recovery, particularly when a specific solvent composition was employed to enhance solubility. We then tested whether the transfer of 5-Ph-IAA-treated *Rosa26$^{OsTIR1/+}$;Satb1$^{V-AID/V-AID}$* T cells into recipient mice accelerate the recovery of Satb1$^{V-AID}$ level. Splenic CD4$^+$ T cells were harvested from *Rosa26$^{OsTIR1/+}$;Satb1$^{V-AID/V-AID}$* mice 16 h after the administration of 5-Ph-IAA, and were intravenously injected into *Rag1*-knockout recipient mice (Supplementary Fig. 2b). This experimental maneuver resulted in the complete restoration of Satb1$^{V-AID}$ level in transferred CD4$^+$ T cells 24 h after transplantation, whereas Satb1$^{V-AID}$ levels in non-transferred CD4$^+$ T cells remained low (Supplementary Fig. 2c, d). This result highlights the feasibility of employing cell transfer as an effective strategy to accelerate the recovery of protein levels following the induction of POI degradation in the OsTIR1-AID2 system.

### Cell type- and stage-specific degradation

Since the removal of the LSL sequence in *Rosa26$^{LSL-OsTIR1}$* allele mediated by *Cd4Cre* occurs specifically in T cells, there was no degradation of Satb1$^{V-AID}$ in B lymphocytes following the administration of 5-Ph-IAA (Fig. 2a), confirming cell type-specific degradation in this system. After intraperitoneal ligand administration, successful Satb1$^{Venus}$ degradation was observed in T cells within the spleen and lymph nodes, as well as in CD8$^+$ T cells in the lungs and T cells in small intestine lamina

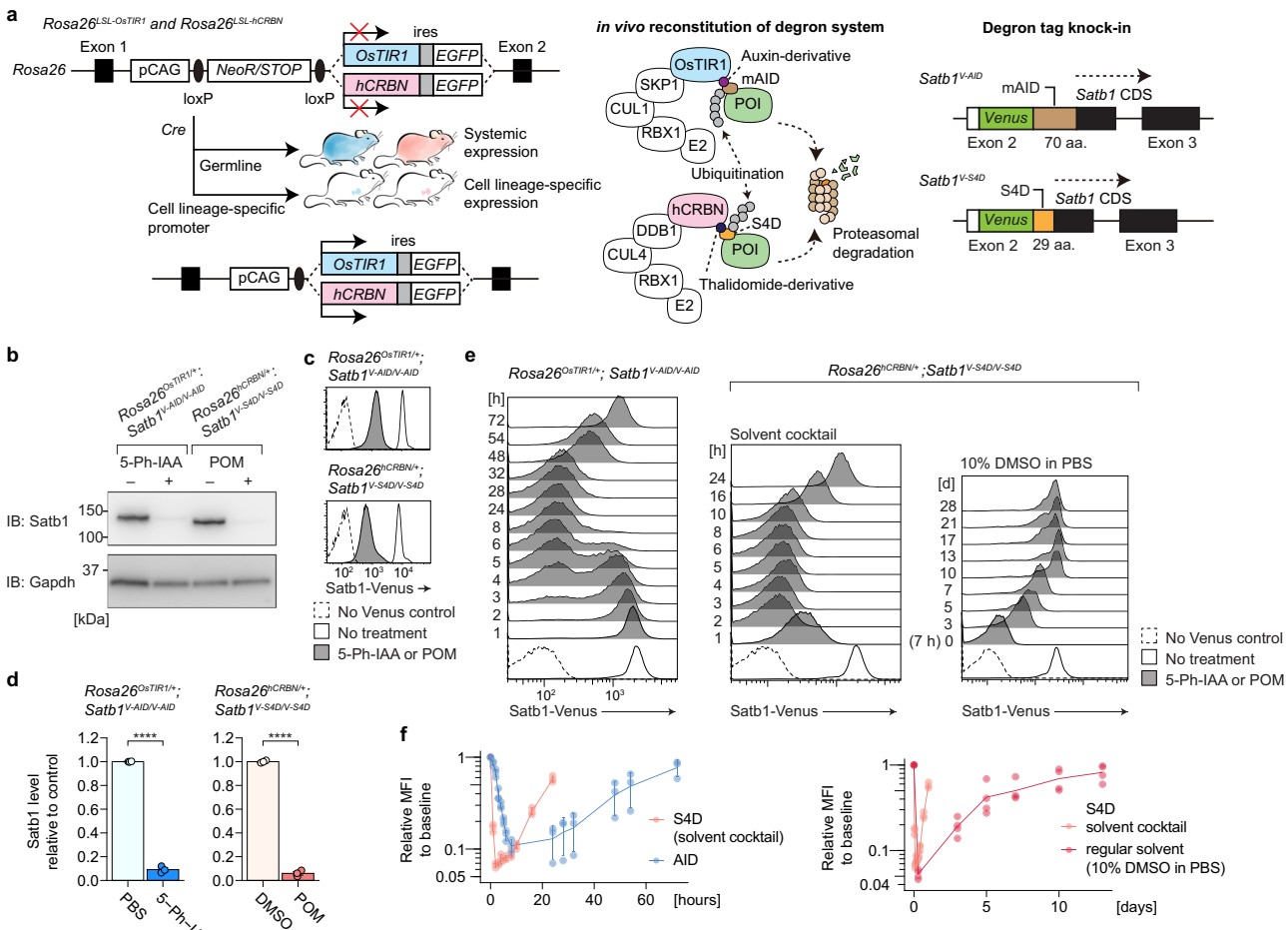

**Fig. 1 | Establishment of in vivo hCRBN-S4D and OsTIR1-AID2 systems in mice.**
**a** Schematic illustration of the hCRBN-S4D and OsTIR1-AID2 systems in mice. The
*OsTIR1*[F74G] and *hCRBN* transgenes are expressed from the *Rosa26* locus, under the
control of the CAG promoter, and upon loxP-Stop-loxP cassette excision by Cre
recombinase. *Satb1*[V-AID] and *Satb1*[V-S4D] were engineered by insertion of mAID
and S4D sequences, respectively, at the junction between Venus and *Satb1* exon 2 in the
*Satb1*[Venus] allele. Immunoblot analysis in total thymocytes (**b**) and flow cytometric
analysis in DP thymocytes (**c**, **d**) after 16 h of intraperitoneal injection with the
indicated ligands. Immunoblot and flow cytometric data are representative of three
independent biological replicates. Graphs in (**d**) are summarized from three inde-
pendent biological replicates ($p = 5.373e-07$ for PBS vs 5-Ph-IAA, $p = 4.743e-07$

for DMSO vs POM). **e**, **f** Kinetics of Satb1 degradation in peripheral blood T cells
post-single dose intraperitoneal administration of 5-Ph-IAA or POM, using a stan-
dard solvent or a solvent cocktail for improved solubility. Histogram data are
representative of three independent biological replicates for OsTIR1-AID2 and
hCRBN-S4D (solvent cocktail), and four independent biological replicates for
hCRBN-S4D (10% DMSO in PBS). Graphs in (**f**) are summarized from three inde-
pendent biological replicates for OsTIR1-AID2 and hCRBN-S4D (solvent cocktail),
and four independent biological replicates for hCRBN-S4D (10% DMSO in PBS).
Data are mean ± s.e.m. Statistical analysis was performed using two-sided unpaired
*t*-test. ****$p < 0.00005$. Source data are provided as a Source Data file. aa. amino
acids, CDS coding sequence, POI protein-of-interest.

propria (Fig. 2b and Supplementary Fig. 3a, b). In addition to the
intraperitoneal injection, oral gavage administration of POM and 5-Ph-
IAA also induced Satb1 degradation in peripheral blood T cells, though
at varying degrees (Supplementary Fig. 3c). Oral administration of 5-
Ph-IAA in *Rosa26*[OsTIR1-ΔEGFP/+];*Satb1*[V-AID/V-AID] mice efficiently decreased the
Satb1[V-AID] levels, down to 3.5% of baseline, whereas orally administered
POM in *Rosa26*[hCRBN/+];*Satb1*[V-S4D/V-S4D] mice showed less efficiency in
reducing Satb1[V-S4D] levels, falling down to only 46.2% of baseline. We
then tested whether these ligands penetrated the blood–brain barrier
and induced the degradation of Satb1 in T cells within the brain par-
enchyma. Satb1[V-AID] and Satb1[V-S4D] protein levels in brain T cells
underwent significant reduction after 16 h post-ligand administration,
although the extent of degradation was slightly lower in brain com-
pared to spleen (Fig. 2b). Notably, the reduction in Satb1 levels within
the brain tended to be greater in the hCRBN-S4D system, although the
difference was not statistically significant. One way to overcome the
low degradation efficacy in brain T cells is to use other IAA or thali-
domide derivatives. In the OsTIR1-AID2 system, 5-adamantyl-IAA (5-Ad-

IAA) was shown to have a higher affinity to OsTIR1[F74G] than 5-Ph-IAA[17].
However, 5-Ad-IAA administration did not induce any significant
degradation of Satb1[V-AID] in splenic and brain T cells (Supplemen-
tary Fig. 3d) due to rapid clearance of 5-Ad-IAA from plasma (Supplemen-
tary Fig. 3e). The other four IAA derivatives tested also failed to show
superior degradation of Satb1 compared with 5-Ph-IAA (Supplemen-
tary Fig. 3f).

We also investigated the potential of the OsTIR1-AID2 and hCRBN-
S4D systems to induce POI degradation in embryos and neonates by
administering the ligands to pregnant or lactating mice, respectively.
One day following intraperitoneal injection of 5-Ph-IAA or POM, a
reduction in Satb1 expression was observed exclusively in *Rosa26*[OsTIR1/+];
*Satb1*[V-AID/V-AID] and *Rosa26*[hCRBN/+];*Satb1*[V-S4D/V-S4D] fetuses, with levels reaching
2.5% and 6.2% of their baseline levels, respectively (Fig. 2c, d). Similarly
to adult mice receiving a single intraperitoneal administration of 5-Ph-
IAA, Satb1[V-AID] levels in *Rosa26*[OsTIR1/+];*Satb1*[V-AID/V-AID] fetuses also recovered
within 3 days post-single intraperitoneal administration in the pregnant
mouse (Supplementary Fig. 3g). One day after ligand administered to

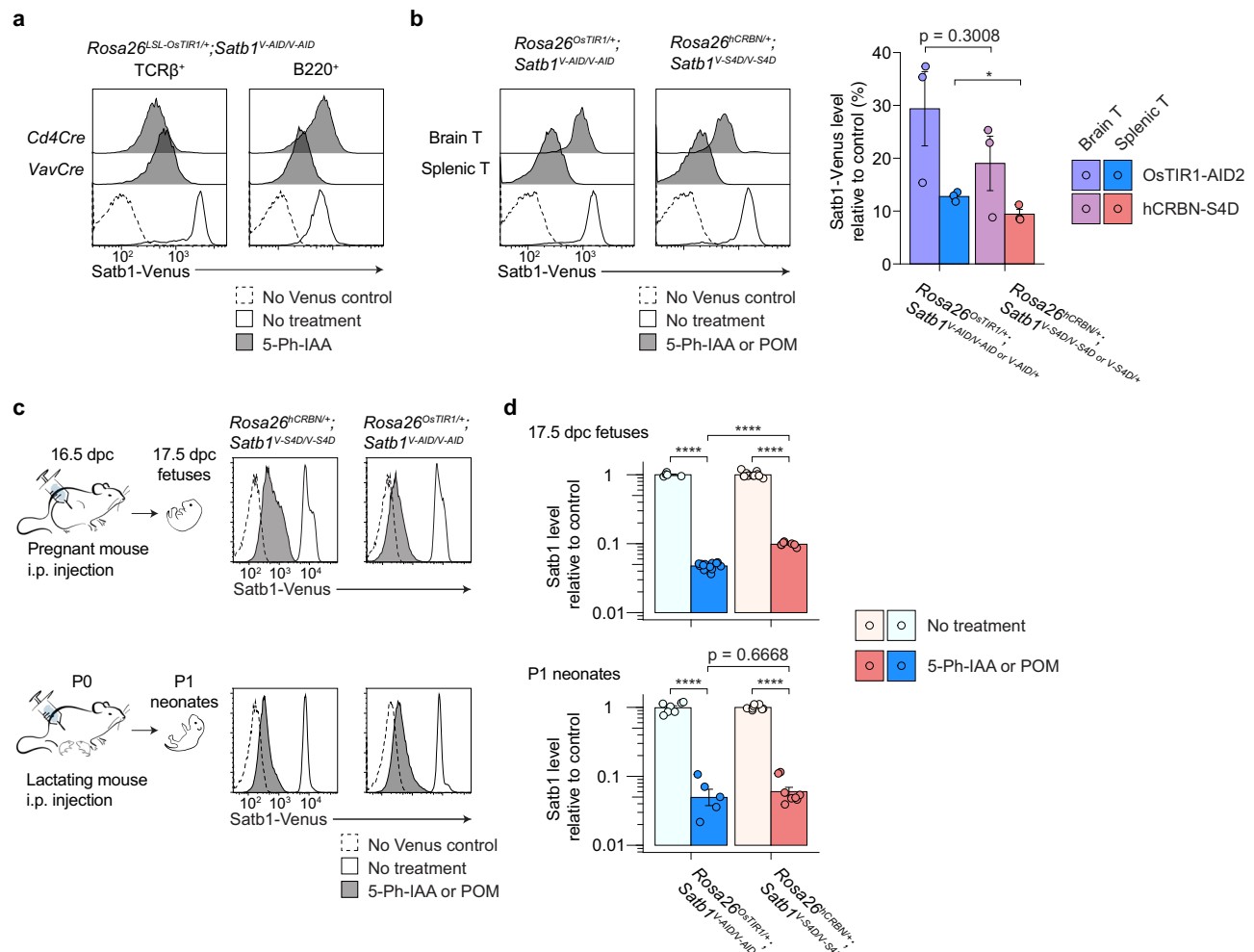

**Fig. 2 | Cell type-specific and developmental stage-specific degradation of Satb1 by degron systems. a** Satb1[Venus] levels in splenic T cells (TCRβ[+]) and B cells (B220[+]), 16 h post-intraperitoneal administration of 5-Ph-IAA. Data are representative of three independent experiments. **b** Satb1[Venus] levels evaluated in T cells from the brain and spleen, 16 h post-intraperitoneal administration of the ligands. Histogram data are representative of three independent experiments and the graph is summarized from three independent experiments ($p = 0.3008$ for brain T cells, $p = 0.03205$ for splenic T cells). **c** Pregnant and lactating mice were treated with intraperitoneal (i.p.) injections of 5-Ph-IAA and POM. Satb1[Venus] levels in fetal and neonatal DP thymocytes were evaluated on the next day. Data are representative of two independent experiments. **d** Summary of Satb1[Venus] level in DP thymocytes from fetuses and neonates, highlighting more pronounced fetal degradation efficacy in the OsTIR1-AID2 system ($n = 5$ and 13 for no-treatment and treatment groups for

17.5 days post coitum (dpc) fetus-data in the OsTIR1-AID2 system, respectively; $n = 13$ and 7 for no-treatment and treatment groups for 17.5 dpc fetus-data in the hCRBN-S4D system, respectively; $n = 7$ and 5 for no-treatment and treatment groups for postnatal day 1 (P1) neonate-data in the OsTIR1-AID2 system, respectively; $n = 8$ and 8 for no-treatment and treatment groups for P1 neonate-data in the hCRBN-S4D system, respectively; $p = <2.2e - 16$ and $2.439e - 16$ for fetal no-treatment vs treatment groups of the OsTIR1-AID2 and hCRBN-S4D systems, respectively; $p = 2.879e - 07$ and $2.087e - 14$ for neonatal no-treatment vs treatment groups of the OsTIR1-AID2 and hCRBN-S4D systems, respectively; $p = 1.743e - 13$ for fetal OsTIR1-AID2 vs hCRBN-S4D treatment groups; $p = 0.6668$ for neonatal OsTIR1-AID2 vs hCRBN-S4D treatment groups. Data are mean ± s.e.m. Statistical analysis was performed using two-sided unpaired $t$-test. *$p < 0.05$, ****$p < 0.00005$. Source data are provided as a Source Data file.

lactating mice, we observed that Satb1[Venus] levels in DP thymocytes of neonates at postnatal day 1 (P1) carrying *Rosa26[OsTIR1/+];Satb1[V-AID/V-AID]* and *Rosa26[hCRBN +];Satb1[V-S4D/V-S4D]* genotypes were efficiently reduced to 3.9% and 3.2% of their baseline levels, respectively (Fig. 2c, d). Varying the dose administration of 5-Ph-IAA in lactating mice resulted in differential levels of Satb1[V-AID] in DP thymocytes of neonatal *Rosa26[OsTIR1/+]; Satb1[V-AID/V-AID]* mice (Supplementary Fig. 3h). Increasing the dose did not provide any significant benefit for Satb1[V-AID] depletion in neonates, suggesting that the standard dosing (0.2 mL of 0.5 mg/mL 5-Ph-IAA) is sufficient for investigating POI depletion during the neonatal period. These findings highlight the versatility of both the OsTIR1-AID2 and hCRBN-S4D systems in inducing POI degradation in embryos and neonates through transplacental circulation and breast milk, thereby supporting the broad applicability of these degron systems across various developmental stages.

## Acute depletion of Satb1 mimics knockout

Next, we investigated the in vivo effects of acute Satb1 depletion on immune phenotypes in mice, specifically focusing on the de-repression of *Foxp3*, which is a known Satb1-deficient phenotype observed in *Satb1[Flox/Fdlox];Cd4Cre* mice[15]. In both *Rosa26[OsTIR1/+];Satb1[V-AID/V-AID]* and *Rosa26[hCRBN/+];Satb1[V-S4D/V-S4D]* mice, 3 days of administrations of 5-Ph-IAA and POM led to a notable increase in the fraction of total Foxp3[+] and CD25[−]Foxp3[+] cells within mature CD4 SP thymocytes and splenic CD4[+] T cells (Fig. 3a, b and Supplementary Fig. 4a). Furthermore, the CD25[−]Foxp3[+] fraction in mature CD8 SP thymocytes increased in 5-Ph-IAA- and POM-treated groups (Fig. 3a and Supplementary Fig. 4a). Conversely, POM-treated *Rosa26[hCRBN/+];Satb1[+/+]* mice did not exhibit these phenotypes, thereby confirming that *Foxp3* de-repression is Satb1 depletion-dependent (Supplementary Fig. 4b). We observed that the frequency of Foxp3[+] cells started to increase from 1 day post-

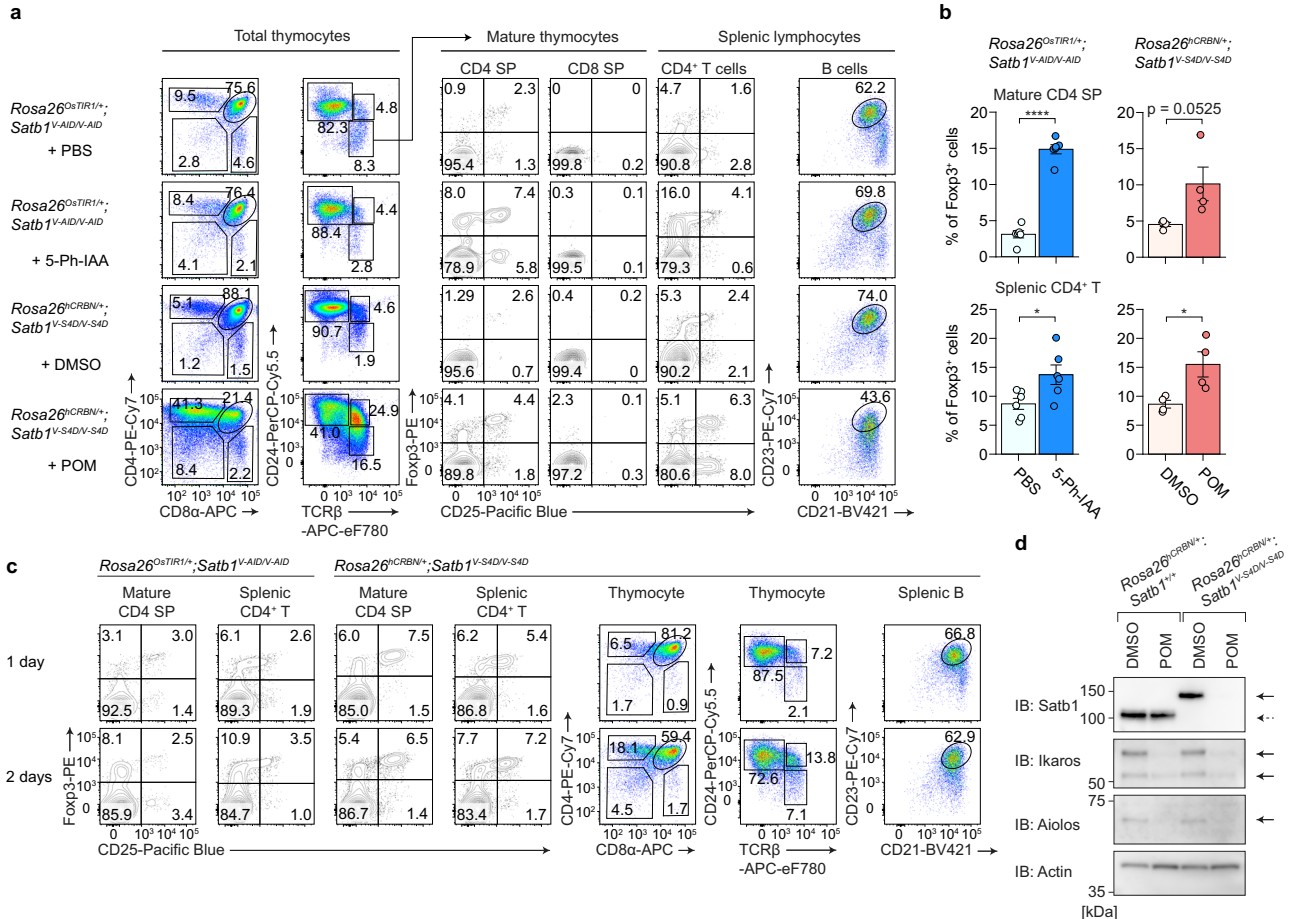

**Fig. 3 | Satb1-deficient phenotype kinetics by acute depletion of Satb1.** Flow cytometric analysis of thymocytes and splenic lymphocytes (**a**) and summary of Foxp3[+] cell fractions within mature CD4 SP thymocytes and splenic CD4[+] T cells (**b**) after 3 consecutive days of 5-Ph-IAA and POM treatment. Flow cytometry data are representative of at least four independent biological replicates (*n* = 6 each for PBS- and 5-Ph-IAA-group in the OsTIR1-AID2 system, respectively; *n* = 4 each for DMSO- and POM-group in the hCRBN-S4D system, respectively), as summarized in (**b**) (*p* = 4.499e − 08 and 0.02642 for PBS vs 5-Ph-IAA in mature CD4 SP and splenic CD4[+] T cells, respectively; *p* = 0.0223 and F = 5.699 for splenic CD4[+] T cells in the hCRBN-S4D system, see also Supplementary Fig. 4). **c** Thymocytes and splenic lymphocytes were subjected to flow cytometric analysis after 1–2 days of 5-Ph-IAA and POM treatment. Data are representative of three independent experiments. **d** Immunoblot analysis of Satb1, Ikaros, Aiolos, and Actin in thymocytes post-16-h treatment. Data are representative of three independent experiments. Data are mean ± s.e.m. Statistical analyses were performed using two-sided unpaired *t*-test (the OsTIR1-AID2 system), or two-sided one-way ANOVA followed by multiple comparisons test using Tukey's "Honest Significant Difference" method (the hCRBN-S4D system). **p* < 0.05, *****p* < 0.00005. Source data are provided as a Source Data file.

treatment in both the OsTIR1-AID2 and hCRBN-S4D systems (Fig. 3c). Given that the de-repression of *Foxp3* by Satb1-deficiency occurs at the transcriptional level[15], these observations indicate that the *Foxp3* gene undergoes rapid activation within 24 h following Satb1 degradation.

### Substrate specificity of degron systems
In addition to Foxp3 de-repression, POM-treated *Rosa26[hCRBN/+]; Satb1[V-S4D/V-S4D]* mice exhibited an increased frequency of CD4 SP population in the thymus and a downregulation of CD23 in splenic B cells (Fig. 3a). Remarkably, these phenotypes were also observed in POM-treated *Rosa26[hCRBN/+];Satb1[+/+]* mice, but not in 5-Ph-IAA-treated *Rosa26[OsTIR1/+];Satb1[V-AID/V-AID]* mice (Fig. 3a and Supplementary Fig. 4b). Given an involvement of hCRBN in the degradation of Ikaros zinc finger (IKZF) family proteins in a thalidomide derivatives-dependent manner[18,19], and the consistent observation of increased CD4 SP population and CD23 downregulation in Ikaros-deficient and Aiolos-mutant mice[20–23], these phenotypes are likely caused by the degradation of Ikaros and Aiolos proteins. Immunoblot analysis confirmed reduced levels of Ikaros and Aiolos in thymocytes from POM-treated *Rosa26[hCRBN/+];Satb1[V-S4D/V-S4D]* and *Rosa26[hCRBN/+];Satb1[+/+]* mice (Fig. 3d). A recent study showed that 5-hydroxylation of POM (5-OH-POM) alters

the substrate specificity of CRL4[CRBN], effectively preventing Ikaros degradation in in vitro experiments[24]. We then tested the effects of 5-OH-POM on the Satb1[V-S4D] and Ikaros in *Rosa26[hCRBN/+];Satb1[V-S4D/V-S4D]* mice and found that the reduction in Satb1[V-S4D] expression induced by 5-OH-POM was notably less pronounced than the effect of POM (Supplementary Fig. 4c, d). Surprisingly, despite the milder Satb1 degradation by 5-OH-POM than POM, *Foxp3* de-repression occurred after 3 days of 5-OH-POM treatment in *Rosa26[hCRBN/+];Satb1[V-S4D/V-S4D]* mice (Supplementary Fig. 4a, e, f). Compared with POM treatment, the thymocytes of *Rosa26[hCRBN/+];Satb1[V-S4D/V-S4D]* mice treated with 5-OH-POM exhibited lower degrees of Ikaros and Aiolos degradation (Supplementary Fig. 4d). Accordingly, an increased frequency of CD4 SP thymocytes and the downregulation of CD23 in splenic B cells were not observed in 5-OH-POM-treated *Rosa26[hCRBN/+];Satb1[V-S4D/V-S4D]* mice (Supplementary Fig. 4e, g).

To investigate the specificity of protein degradation and identify potential neo-substrates for the OsTIR1-AID2 and hCRBN-S4D systems, we conducted proteomic analyses of thymocytes 16 h after ligand injection. Among the ~6500 proteins detected, 6 proteins were decreased in the thymus of 5-Ph-IAA-treated *Rosa26[OsTIR1/+];Satb1[V-AID/V-AID]* mice compared to those in PBS-treated control mice, with an adjusted *p* value of <0.0005 (Fig. 4a–c and Supplementary Fig. 5a). In contrast,

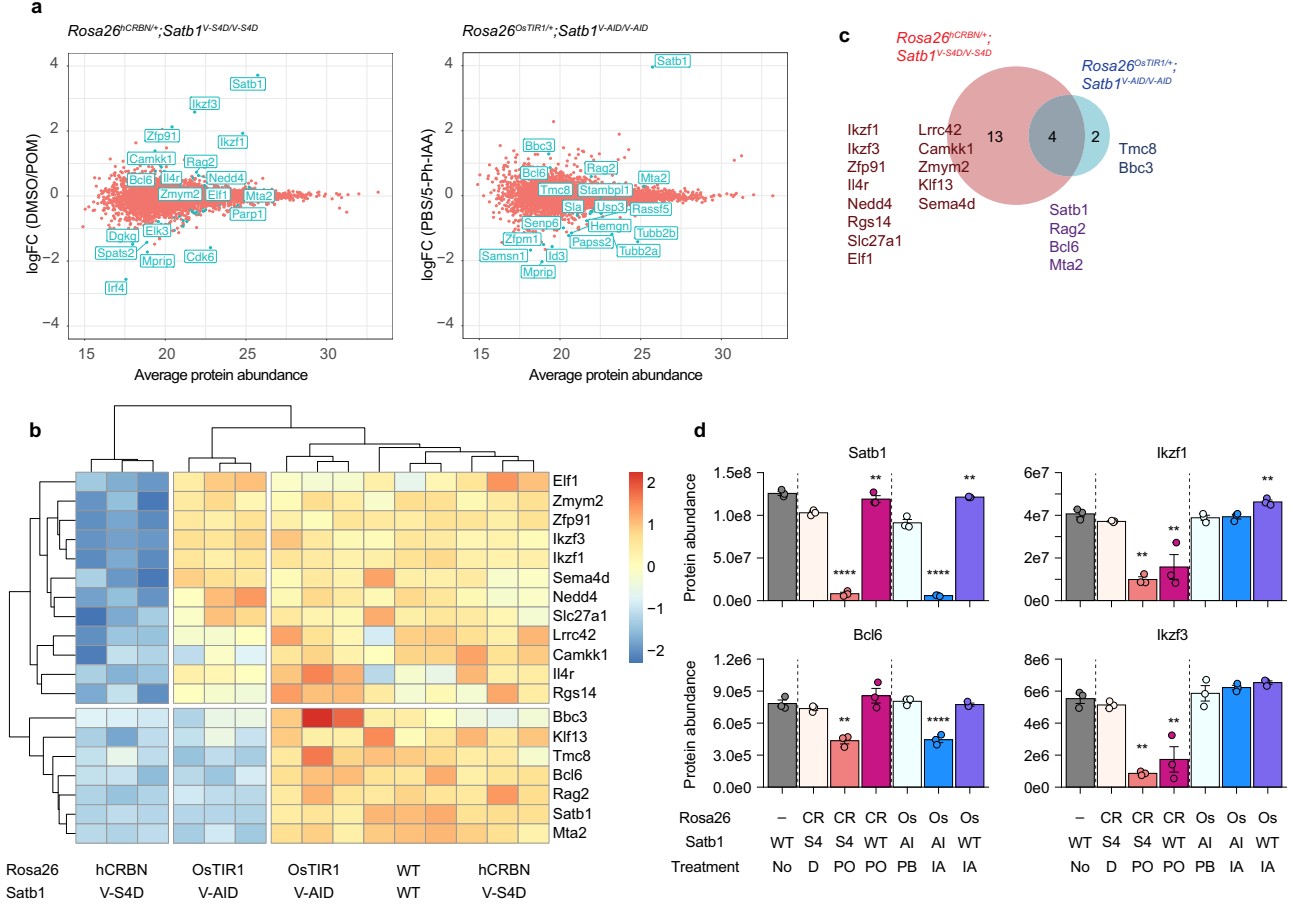

**Fig. 4 | Substrate specificity of the OsTIR1-AID2 and hCRBN-S4D systems.**
**a** Summary of protein abundance and fold changes detected by LC−MS in thymus of indicated genotypes and treatments (*n* = 3 for each condition). Proteins showing significant changes with adjusted *p* value < 0.0005 are highlighted in green.
**b** Heatmap displaying protein abundances across different genotypes and treatments. All proteins that were significantly reduced upon 5-Ph-IAA and POM treatment (with adjusted *p* value < 0.0005) in the thymus of *Rosa26^OsTIR1/+^;Satb1^V-AID/V-AID^* and *Rosa26^hCRBN/+^;Satb1^V-S4D/V-S4D^* mice are included. Values were normalized to the mean abundance among samples for each protein. **c** Venn diagram illustrating shared proteins with significant decrease in abundance across both treatment groups. **d** The abundances of Satb1, Bcl6, Ikaros (Ikzf1), and Aiolos (Ikzf3) in the thymus of indicated genotypes and treatments. Treatment groups were compared to their respective controls (DMSO-treated *Rosa26^hCRBN/+^;Satb1^V-S4D/V-S4D^* for the

hCRBN-S4D system and PBS-treated *Rosa26^OsTIR1/+^;Satb1^V-AID/V-AID^* for the OsTIR1-AID2 system). $P = 7.79e-08$ and $F = 699.5$ for Satb1 in the OsTIR1-AID2 system; $p = 2.29e-07$ and $F = 487.3$ for Satb1 in the hCRBN-S4D system; $p = 3.04e-05$ and $F = 93.13$ for Bcl6 in the OsTIR1-AID2 system; $p = 0.00138$ and $F = 23.92$ for Bcl6 in the hCRBN-S4D system; $p = 0.00458$ and $F = 15.07$ for Ikzf1 in the OsTIR1-AID2 system; $p = 0.00318$ and $F = 17.39$ for Ikzf1 in the hCRBN-S4D system; $p = 0.345$ and $F = 1.277$ for Ikzf3 in the OsTIR1-AID2 system; $p = 0.00147$ and $F = 17.39$ for Ikzf1 in the hCRBN-S4D system. *Y* axis represents relative protein abundance. Data are mean ± s.e.m. Statistical analyses were performed using two-sided empirical Bayes moderated *t*-statistics (**a**), or two-sided one-way ANOVA followed by Dunnett's multiple comparisons test (**d**). ***p* < 0.005, *****p* < 0.00005. Source data are provided as a Source Data file. CR, *Rosa26^hCRBN/+^*; Os, *Rosa26^OsTIR1/+^*; WT, *Satb1^+/+^*; S4, *Satb1^V-S4D/V-S4D^*; AI, *Satb1^V-AID/V-AID^*; No, no treatment; PO, POM; IA, 5-Ph-IAA.

17 proteins were decreased in the thymocytes of POM-treated *Rosa26^hCRBN/+^;Satb1^V-S4D/V-S4D^* mice compared to those treated with DMSO, with an adjusted *p* value of <0.0005 (Fig. 4a–c and Supplementary Fig. 5b). Among these, Satb1, Rag2, Bcl6, and Mta2 were commonly reduced in POM-treated *Rosa26^hCRBN/+^;Satb1^V-S4D/V-S4D^* and 5-Ph-IAA-treated *Rosa26^OsTIR1/+^;Satb1^V-AID/V-AID^* mice (Fig. 4b–d and Supplementary Fig. 5c). Thirteen proteins were decreased specifically in POM-treated *Rosa26^hCRBN/+^;Satb1^V-S4D/V-S4D^* mice, but not in 5-Ph-IAA-treated *Rosa26^OsTIR1/+^;Satb1^V-AID/V-AID^* mice. As expected, known neo-substrates, such as Ikaros, Aiolos, Zfp91[25], and Zmym2[26], were present in this group (Fig. 4b–d and Supplementary Fig. 5c). Only Tmc8 and Bbc3 were specifically decreased in 5-Ph-IAA-treated *Rosa26^OsTIR1/+^;Satb1^V-AID/V-AID^* mice and not in POM-treated *Rosa26^hCRBN/+^;Satb1^V-S4D/V-S4D^* mice (Fig. 4c). These proteins also exhibited a tendency to decrease in the hCRBN-S4D system (Tmc8, logFC 0.23 with adjusted *p* value 0.04; Bbc3, logFC 0.30 with adjusted *p* value 0.32), raising the possibility of secondary effects of Satb1 degradation. Our proteomics analysis of the

cerebrum of DMSO- and POM-treated *Rosa26^hCRBN/+^;Satb1^+/+^* mice did not detect any proteins with statistically significant differences in abundance among the ~7000 proteins detected (Supplementary Fig. 5d). This suggests that the hCRBN-S4D system has the potential to serve as a specific TPD technology for studying the temporal depletion of POI in the brain.

## Degradation of plasma membrane proteins

Having successfully established in vivo degron systems capable of effectively degrading Satb1, we next explored the feasibility of targeting plasma membrane proteins for degradation using degron systems. We selected programmed cell death-1 (PD-1) and a G protein-coupled receptor CXCR4 as targets. We expressed PD-1 and CXCR4 tagged with mAID in their cytoplasmic tails, along with OsTIR1^F74G^ in Jurkat T and HEK293T cell lines, respectively, and supplemented the cultures with 1 μM of 5-Ph-IAA. Within 4 and 6 h, the fraction of cells expressing PD-1-mAID and CXCR4-mAID on their cell surface was

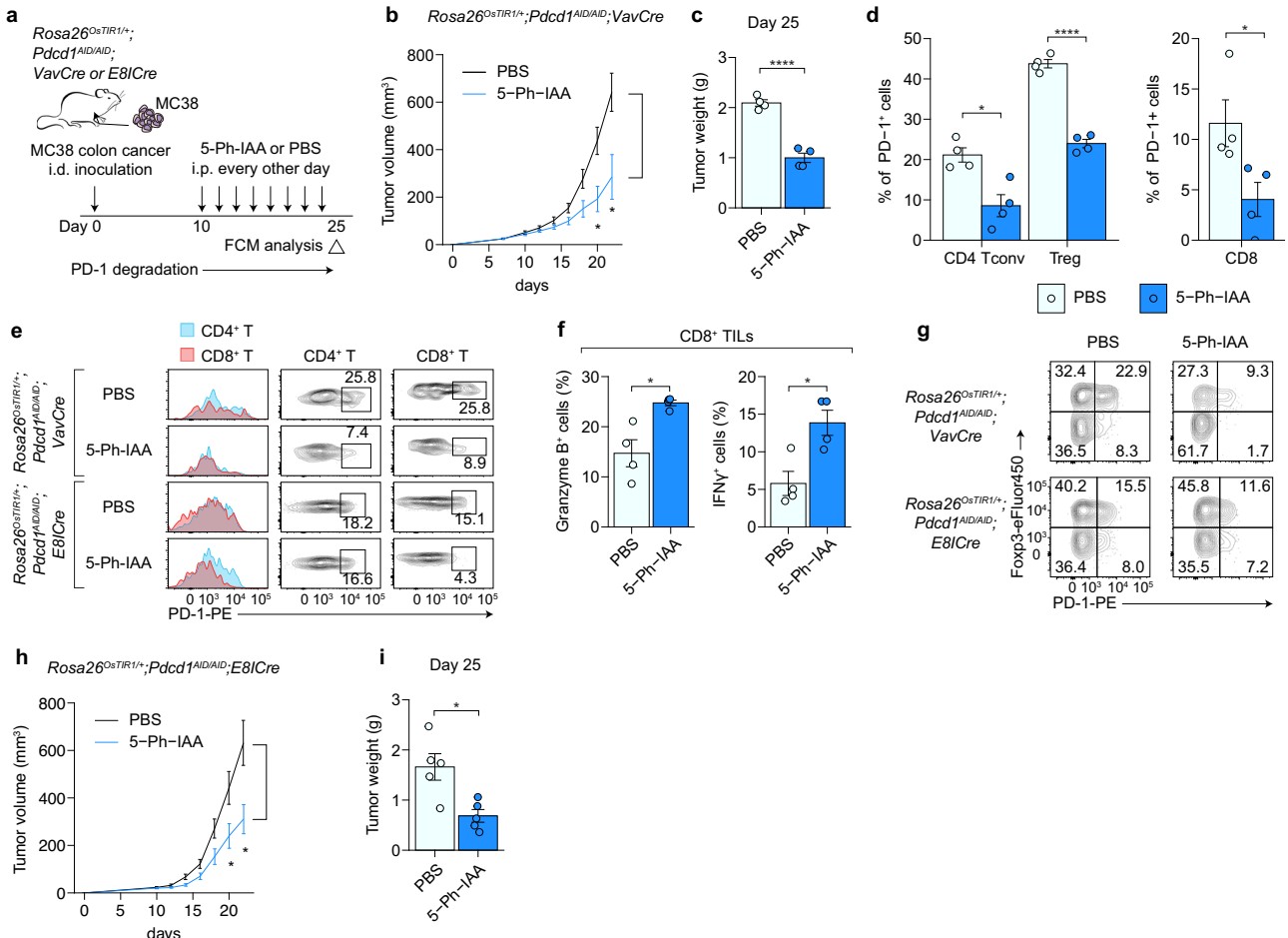

**Fig. 5 | PD-1 degradation using the OsTIR1-AID2 system. a–c, h, i** Anti-tumor effect of temporal PD-1 degradation in hematopoietic cells. MC38-inoculated *Rosa26^(LSL-OsTIR1/+)*;*Pdcd1^(AID/AID)*;*VavCre* (**b**, **c**) or *Rosa26^(LSL-OsTIR1/+)*;*Pdcd1^(AID/AID)*;*E8ICre* (**h**, **i**) mice were intraperitoneally (i.p.) injected with PBS or 5-Ph-IAA every 2 days, starting from 10 days after tumor inoculation. Tumor volume was assessed at specified time points (**b**, **h**; *n* = 7 for PBS group and *n* = 6 for 5-Ph-IAA group for *VavCre*; *n* = 11 for PBS group and *n* = 10 for 5-Ph-IAA group; *p* = 0.0104 and 0.01433 for day 20 and 22 for *VavCre*; *p* = 0.03241 and 0.01181 for *E8ICre*), and tumor weight was measured after 25 days, post-tumor inoculation (**c**, **i**; *n* = 4 each for *VavCre* and *n* = 5 each for *E8ICre*; *p* = 7.028e − 05 for *VavCre* and *p* = 0.01037 for *E8ICre*). **d** Proportion of PD-1⁺ cells within CD4⁺ conventional T cells (Tconv, Foxp3⁻) and Treg (Foxp3⁺) cells (left) and CD8⁺ T cells (right) was examined after 25 days of MC38 inoculation in *Rosa26^(LSL-OsTIR1/+)*;*Pdcd1^(AID/AID)*;*VavCre* mice (*n* = 4 for each group, *p* = 0.008301, 1.033e − 05, and 0.03902 for CD4⁺ Tconv, Treg, and CD8⁺ T cells,

respectively). **e** PD-1 expression in CD4⁺ and CD8⁺ T cells within tumor was examined after 25 days post-MC38 inoculation. Data are representative of *n* = 4 each (PBS and 5-Ph-IAA) for *Rosa26^(LSL-OsTIR1/+)*;*Pdcd1^(AID/AID)*;*VavCre* and *n* = 5 each (PBS and 5-Ph-IAA) for *Rosa26^(LSL-OsTIR1/+)*;*Pdcd1^(AID/AID)*;*E8ICre* mice. **f** Fraction of Granzyme B⁺ and IFNγ⁺ cells within CD8+ tumor-infiltrating lymphocytes (TILs) in *Rosa26^(LSL-OsTIR1/+)*;*Pdcd1^(AID/AID)*;*VavCre* mice 25 days, post-MC38 inoculation (*n* = 4 for each group, *p* = 0.01073 and 0.01312 for Granzyme B and IFNγ, respectively). **g** PD-1 and Foxp3 expression in CD4⁺ T cells within tumor was examined after 25 days, post-MC38 inoculation. Data are representative of *n* = 4 each (PBS and 5-Ph-IAA) for *Rosa26^(LSL-OsTIR1/+)*;*Pdcd1^(AID/AID)*;*VavCre* and *n* = 5 each (PBS and 5-Ph-IAA) for *Rosa26^(LSL-OsTIR1/+)*;*Pdcd1^(AID/AID)*;*E8ICre* mice. Data are mean ± s.e.m. Statistical analysis was performed using two-sided unpaired *t*-test. **p* < 0.05, *****p* < 0.00005. Source data are provided as a Source Data file.

significantly reduced (Supplementary Fig. 6a), indicating that the OsTIR1-AID2 system effectively induces the degradation of plasma membrane proteins. Notably, the fusion of the mAID tag to the extracellular domain did not effectively degrade PD-1 (Supplementary Fig. 6b). These results highlight the importance of selecting the intracellular domain as the fusion site for the degron tag when plasma membrane proteins are targeted for degradation.

To assess the in vivo application of PD-1 degradation by the OsTIR1-AID2 system, we inserted an mAID degron sequence with a linker peptide sequence into the C-terminal coding end of endogenous murine *Pdcd1* gene by genome editing, thereby generating *Pdcd1^(AID)* allele (Supplementary Fig. 6c). In *Rosa26^(LSL-OsTIR1/+)*;*Pdcd1^(AID/AID)*;*VavCre* mice, the expression of PD-1-mAID on follicular helper T cells was completely lost within 4–6 h after the intraperitoneal injection of 5-Ph-IAA and returned to untreated baseline levels within 3 days (Supplementary Fig. 6d). Notably, the extent of PD-1-mAID degradation

appeared to be higher than that of a recently reported small molecule-assisted shutoff (SMASh) degron system[27]. Subsequently, we examined the impact of transient PD-1 depletion on the anti-tumor effect using the colon cancer cell line MC38 xenograft model (Fig. 5a). Mice treated with 5-Ph-IAA exhibited smaller tumor volume compared to PBS-treated controls at 20 days post-tumor inoculation and beyond (Fig. 5b, c). A substantial reduction in PD-1-mAID level in T cells was observed within tumor-infiltrating lymphocytes (TILs) in 5-Ph-IAA-treated mice compared to PBS-treated controls (Fig. 5d, e). Compared to PBS-treated controls, the frequency of Granzyme B⁺ and IFN-γ⁺ CD8⁺ effector TILs was increased in 5-Ph-IAA-treated mice. (Fig. 5f). Interestingly, Foxp3⁺ regulatory T cells (Tregs) displayed a relatively spared PD-1⁺ cell fraction within CD4⁺ TILs in 5-Ph-IAA-treated mice (Fig. 5d, g). Subsequently, we attempted to elucidate the impact of PD-1 degradation specifically in CD8⁺ T cells by using *Rosa26^(LSL-OsTIR1/+)*;*Pdcd1^(AID/AID)*;*E8ICre* mice, where Cre recombinase is exclusively

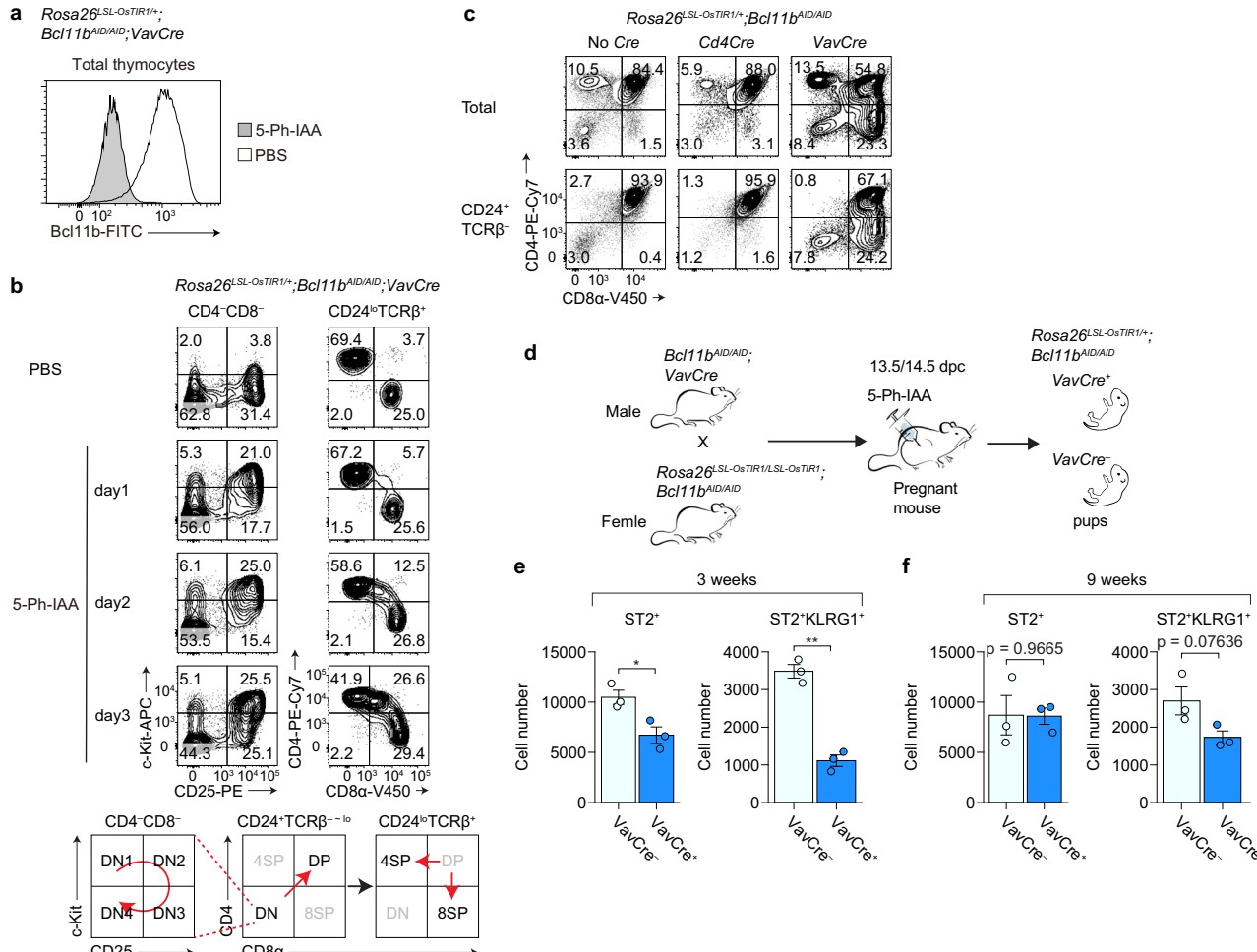

**Fig. 6 | Bcl11b degradation by the OsTIR1-AID2 system. a** Expression of Bcl11b in total thymocytes analyzed by flow cytometry 16 h after a single-dose intraperitoneal injection of PBS or 5-Ph-IAA. Data are representative of three independent experiments. **b** Thymocytes were analyzed by flow cytometry after 1, 2, and 3 days of daily 5-Ph-IAA treatment or 3-day treatment of PBS. Data are representative of three independent experiments. A schematic diagram for gating the indicated thymocyte populations is shown at the bottom. **c** Flow cytometric analysis of thymocytes in mice treated with 5-Ph-IAA for 3 days. Data are representative of three

independent experiments. **d–f** Pregnant *Rosa26[LSL-OsTIR1/ LSL-OsTIR1];Bcl11b[AID/AID]* female mice mated with *Bcl11b[AID/AID];VavCre* male mice received intraperitoneal injection of 5-Ph-IAA at 13.5 and 14.5 days of gestation (**d**). The total lung ILC2 (CD45+CD3−Thy1.2+ST2+) numbers were examined in the offspring at 3 weeks (**e**) and 9 weeks (**f**), by flow cytometry (*n* = 3 for each group; *p* = 0.02488 and 0.0005357 for ST2+ and ST2+KLRG1+ ILC2 at 3 weeks). Data are mean ± s.e.m. Statistical analysis was performed using two-sided unpaired *t*-test. *$p < 0.05$, **$p < 0.005$. Source data are provided as a Source Data file.

expressed in CD8+ T cells[28]. This selective PD-1 degradation in CD8+ T cells resulted in an anti-tumor response, comparable to that observed in *Rosa26[LSL-OsTIR1/+];Pdcd1[AID/AID];VavCre* mice in the MC38 xenograft model (Fig. 5h, i and Supplementary Fig. 6e), thereby implying that the therapeutic anti-tumor effects associated with anti-PD-1 therapy predominantly stem from the inhibition of PD-1 signaling within CD8+ T cells. These observations highlight a significant advantage of degron systems when investigating the cell type-specific effects of the temporal loss of plasma membrane protein expression, which is a feature that cannot be achieved through systemic administration of neutralizing and blocking antibodies.

## Bcl11b depletion impacting thymocyte and ILC2 development

Bcl11b is a transcription factor that plays an essential role in T-lymphocyte development[29–31]. Currently available technologies do not allow us to study the role of Bcl11b in the transition from CD4−CD8− double-negative (DN) to CD4+CD8+ DP stage, primarily because of the lack of appropriate Cre-transgenic mice. By introducing the mAID sequence into the N-terminal *Bcl11b*-coding exon through gene editing, we generated a *Bcl11b[AID]* allele that produces a Bcl11b-mAID

fusion protein (Supplementary Fig. 7a). 5-Ph-IAA administration to *Rosa26[LSL-OsTIR1/+];Bcl11b[AID/AID];VavCre* mice effectively degraded Bcl11b-mAID protein in 16 h (Fig. 6a). The mice were then subjected to daily intraperitoneal injections of either PBS or 5-Ph-IAA for 3 consecutive days. We observed an accumulation of c-Kit+CD25+ DN2 thymocytes, a phenotype that results from the loss of Bcl11b[30], as early as 24 h after a single dose of 5-Ph-IAA (Fig. 6b). In contrast, aberrant mature (CD24loTCRβ+) CD4+CD8+ DP thymocytes emerged after 2 consecutive days of treatment (Fig. 6b). The different kinetics of phenotypic appearance suggest that the transition from the DN2 to the DN3 stage is a rapid process, while the development of mature thymocytes from immature CD4+CD8+ DP precursors takes longer. Interestingly, the frequency of CD4−CD8+ SP population in total thymocytes increased due to the increase of CD24+ TCRβ− immature CD4−CD8+ thymocytes, and this CD8+ SP population was not observed in 5-Ph-IAA-treated *Rosa26[LSL-OsTIR1/+];Bcl11b[AID/AID];Cd4Cre* mice (Fig. 6c).

Previous studies have demonstrated that the inactivation of *Bcl11b* in innate lymphoid cell (ILC) progenitors during embryonic development inhibits the generation of group 2 ILC (ILC2s)[32,33]. In adult mice, ILC2s in various tissues, particularly lung ILC2, have been shown

to acquire tissue residence and are not replaced by circulating ILC2s or ILC progenitors in a steady state[34]. Given the capability of degron systems to induce temporal protein loss in embryos, we aimed to investigate whether and to what extent the temporal loss of Bcl11b during embryogenesis inhibits lung ILC2 generation and whether the restoration of Bcl11b expression can reestablish lung ILC2 homeostasis. To address this, we intraperitoneally administered 5-Ph-IAA to pregnant mice bearing $Rosa26^{LSL-OsTIR1/+}$;$Bcl11b^{AID/AID}$;$VavCre^{+ or -}$ fetuses at 13.5 and 14.5 days post coitum (dpc) (Fig. 6d). At 3 weeks of age, the $VavCre^+$ mice that underwent temporal Bcl11b loss during embryogenesis exhibited reduced numbers of ILC2s, especially KLRG1$^+$ ILC2, in the lungs (Fig. 6e). Since fetal depletion of Bcl11b from 13.5 dpc to birth did not have an additive effect in reducing lung ILC2s (Supplementary Fig. 7b), we used 13.5–14.5 dpc depletion model for further analyses. Interestingly, at 9 weeks of age, there were no significant differences in the number of lung ILC2s between the $VavCre^-$ and $VavCre^+$ groups (Fig. 6f). These observations suggested that the reduced number of lung ILC2, which stems from the loss of Bcl11b expression during embryogenesis, leaves the lung niche open for bone marrow-derived or tissue-resident ILC progenitors to replenish ILC2s in adults[35].

## Discussion

We developed two in vivo degron systems, the OsTIR1-AID2 and hCRBN-S4D systems, for the inducible, rapid, and temporal degradation of POI in mice. Using Satb1 as a model POI, we conducted a comparative analysis of the kinetics and the specificity of protein degradation of the two degron systems. The OsTIR1-AID2 system demonstrated a higher degree of specificity than the hCRBN-S4D system. Alongside Satb1, Rag2, and Bcl6 exhibited reduced levels in both degron systems. Given that Satb1 binds to the super-enhancer regions near the $Rag2$ and $Bcl6$ loci, positively regulating the expression of these genes[36,37], decrease in Rag2 and Bcl6 protein levels is likely a consequence of a reduction in transcription through the attenuated activity of these enhancers upon loss of Satb1. Mta2 also exhibited reduced levels in both degron systems. Based on previous studies reporting protein interaction between Satb1 and Mta2[38], the degradation of Satb1 may have likely contributed to the instability of Mta2 proteins. Conversely, the hCRBN-S4D system degraded several neo-substrates, such as IKZF family proteins. In line with a previous report demonstrating the reduced activity of 5-OH-POM against IKZF family proteins in vitro[24], degradation of Ikaros and Aiolos was not observed in the hCRBN-S4D system after treatment with 5-OH-POM, although the extent of Satb1 degradation was also less than that of POM treatment. Based on our kinetics study, it is possible to regulate the kinetics of POI degradation by modifying the solubility and bioavailability of the ligands. Extended degradation of Satb1 over 1 week has been achieved using the hCRBN-S4D system, with the standard POM solution in 10% DMSO, PBS. Sustained depletion of POI could be maintained with weekly or bi-weekly administrations, which is beneficial for studying long-term POI depletion. Additionally, when using a solvent cocktail of 15% DMSO, 17.5% Cremophor EL, 8.75% Ethanol, 8.75% HCO-40, and 50% PBS for POM solution, the hCRBN-S4D system demonstrated sharper kinetics in POI depletion, compared to the OsTIR1-AID2 system. Considering the high specificity of target protein degradation, the OsTIR1-AID2 system is currently a more desirable choice for basic research. Further optimization, including the modification of OsTIR1 which functions even at lower concentrations of auxin derivatives, could substantially expand the versatility of the OsTIR1-AID2 system, opening up possibilities for a broader range of applications.

The inducible and rapid loss of protein enable us to examine the kinetics of phenotypic changes following the loss of the POI. For instance, we showed that the acute depletion of Satb1 led to substantial dysregulation of the expression of some Satb1 target genes,

such as $Rag2$ and $Bcl6$, within 16 h. Furthermore, compared to irreversible genome engineering technologies including Cre-loxP-mediated conditional gene inactivation, one notable advantage of the degron system is its ability to recover protein expression within the same cells. This enables researchers to investigate how temporal perturbations in POI function over varying time periods influence cell function and phenotypes later in vivo, as exemplified in our Bcl11b study. POI can be degraded in embryos or neonates by ligand injection into pregnant or lactating mice, respectively, thereby making it possible to address how temporal perturbation of protein function in early life influences the risk of disease development in later life. In this study, we revealed that Bcl11b is essential for the efficient differentiation of DN thymocytes into DP thymocytes through the rapid loss of the Bcl11b protein in cells at this transition stage. An increase in immature CD8 SP thymocytes has been observed in mice harboring $Bcl11b^{N797K}$ missense mutation[39], which supports this observation. Consequently, bypassing the limitations of Cre transgene availability in degron systems provides an opportunity that has not–to the best of our knowledge–yet been achieved using the Cre-loxP system.

The application of these degron systems in cells of the central nervous system has significant implications for biomedical research. As the degradation of degron-tagged Satb1$^{Venus}$ in the brain by these degron systems was comparatively less pronounced than that in the spleen and thymus, the development of auxin derivatives with a higher affinity for OsTIR1$^{F74G}$ would enhance blood–brain barrier penetration. In this study, we conducted experiments to verify the efficient degradation of plasma membrane receptors, including G protein-coupled receptors possessing seven transmembrane domains, using the OsTIR1-AID2 system. The temporal loss of cell-surface proteins in mice can serve as a model for simulating the beneficial and adverse effects of neutralizing antibody therapies that target cell-surface proteins. In degron systems, the cell type-specific expression of E3 ubiquitin ligases can be achieved through the selection of appropriate Cre transgenes. This capability allows us to investigate the impact of the loss of protein function in specific cell types, whereas the inherent lack of cell-type specificity is a drawback of systemic antibody administration. In cases of adverse reactions occurring in systemic antibody administration, it is difficult to identify the cells responsible for the adverse effects, due to the systemic effects of the antibody. Immune checkpoint blockade therapy using anti-PD-1 or PD-L1 antibodies in cancer patients has been associated with autoimmune-like adverse effects, although the underlying mechanisms remain poorly understood[40,41]. In the degron system, however, PD-1 degradation can be specifically induced in CD8$^+$ T cells, regulatory T cells and monocyte-lineage cells by using $E8ICre$, $Foxp3Cre$, and $LysMCre$, respectively. Consequently, dissecting the effects of PD-1 degradation in each of these cell types, either individually or in combination, can potentially yield valuable insights into maximizing the effects of immune checkpoint blockade therapy while minimizing adverse effects. Additionally, raising a specific antibody against a G protein-coupled receptor, a process that is both cost- and time-consuming, is not always successful and is fraught with the problem of uncertain specificity. Given that genetic targeting of the degron module to the POI is a quick and reliable method for ensuring the specificity of proteins that lose function upon degradation, the degron system solves the above issues in antibody production, in particularly for the case of therapeutic applications.

Several areas remain unexplored in which the degron system could benefit from further refinement. The degron system requires the tagging of POI with mAID or S4D motifs, and the addition of these degron motifs occasionally reduces the expression level of POI. It should also be noted that the addition of a degron tag to the POI may impact its biological function, as recent studies have reported phenotypic changes caused by the degron-POI fusion[11]. Therefore, optimizing the position of the degron tag in POI, as well as including linker sequences, should be carefully considered, and any resultant

phenotypic changes should be thoroughly investigated. Furthermore, the extent of POI degradation differs across cell types. Although the degree of Satb1 degradation did not differ significantly among the T cell subsets examined, PD-1 degradation seemed to be less pronounced in Tregs than in conventional CD4$^+$ T cells. Therefore, it is important to consider the POI degradation efficiency in each cell type.

In summary, in vivo TPD systems developed in this study enabled us to achieve rapid, efficient, and temporal degradation of intracellular and membrane proteins within selective targeted cell lineages. These systems offer a promising approach for elucidating the functions of target proteins within specific cellular populations, with extensive applications in diverse fields of biological and medical research.

## Methods

### Ethical statement
The experimental protocols for animal studies were approved by the Institutional Animal Care and Use Committee of RIKEN Yokohama Branch (AEY2022-019[2]).

### Mice
B6N.Cg-Commd10$^{Tg(Vav1-icre)A2Kio}$/J (*VavCre*, 018968), C57BL/6-Tg(Cd8a-cre)1Itan/J (*E8ICre*, 008766)[42], B6.FVB-Tg(EIIa-cre)C5379Lmgd/J (*EIIaCre*, 003724) and B6.129S7-Rag1$^{tm1Mom}$/J (*Rag1*-deficient, 002216) mice were purchased from the Jackson laboratory. *Cd4Cre* mice and *Satb1$^{Flox}$* mice were provided by Dr. Christopher B. Wilson and Dr. Terumi Kowhi-Shigematsu, respectively. All mice were maintained at the RIKEN Center for Integrative Medical Sciences. Mice were maintained under specific-pathogen free conditions with controlled temperature and humidity with 12-h dark/light cycle. The experimental protocols for animal studies were approved by the Institutional Animal Care and Use Committee of RIKEN Yokohama Branch (AEY2022-019[2]). 17.5–18.5 dpc fetuses, P1 neonates, 3-week-old, and 6–12-week-old mice were used in the flow cytometric and proteomic analyses. Age-matched animals were used for all experiments.

### Generation of *Rosa26$^{LSL-OsTIR1}$* mouse strain
To generate a *Rosa26$^{LSL-OsTIR1}$* allele, cDNA encoding *OsTIR1$^{F74G}$* was amplified by PCR to introduce AscI sites at the both ends, and inserted into an AscI-cleaved pCTV vector (Addgene, 15912). Subsequently, 30 μg of the target vector was linearized by AsiSI enzyme and was transfected into the M1 embryonic stem (ES) cell line by electroporation as previously described[43]. After G418 selection, ES clones that were G418-resistant were further screened for the homologous recombination event by PCR. Clones that were confirmed to have undergone the desired recombination were aggregated with blastocysts to generate chimera mice, through which the *Rosa26$^{LSL-OsTIR1}$* allele was germline transmitted to the offspring, establishing the *Rosa26$^{LSL-OsTIR1}$* mouse line. In order to establish mouse line harboring the *Rosa26$^{OsTIR1}$* allele without the stop cassette, *Rosa26$^{LSL-OsTIR1}$* mice were crossed with *EIIaCre* line, expressing Cre recombinase in germ cells. Offspring with the excised stop cassette were selected as founder for the *Rosa26$^{OsTIR1}$* line. To generate the *Rosa26$^{OsTIR1-ΔEGFP}$* allele, Cas9 mRNA, tracrRNA, and crRNA, targeting both the N-terminal and C-terminal ends of the EGFP sequence within the *Rosa26$^{OsTIR1}$* allele, were electroporated into *Rosa26$^{OsTIR1/+}$* zygotes, to excise the EGFP sequence from the *Rosa26$^{OsTIR1}$* allele. Founders with successful EGFP excision were selected to establish the *Rosa26$^{OsTIR1-ΔEGFP}$* mouse strain.

### Generation of *Rosa26$^{LSL-hCRBN}$* mouse strain
The transgenic line expressing the hCRBN-AirID (ancestral BirA for proximity-dependent biotin identification) protein from the *Rosa26* locus[44] was first generated via the gene targeting strategy and, as described above, the pCTV vector (Addgene, 15912) was used to ligated the DNA fragment encoding hCRBN-AirID into an AscI-site. After establishing the *Rosa26$^{LSL-hCRBN-AirID}$* transgenic line, a stop codon

between the *hCRBN* and *AirID* coding sequence was introduced, using the CRISPR/Cas9 genome-editing approach, thereby generating a *Rosa26$^{LSL-hCRBN}$* allele. Sequences for guide RNA and ssODNs for this gene editing are listed in Supplementary Table 1. To generate a mouse line harboring a *Rosa26$^{hCRBN}$* allele without the stop cassette, *Rosa26$^{LSL-hCRBN}$* mice were crossed with B6.FVB-Tg(EIIa-cre)C5379Lmgd/J (*EIIaCre;* Jackson laboratory, 003314), and founders that have the excised stop cassette were selected.

### Generation of *Satb1$^{V-S4D}$*, *Satb1$^{V-AID}$*, *Bcl11b$^{AID}$*, and *Pdcd1$^{AID}$* mouse strains
Generation of *Satb1$^{Venus}$* strain was previously described[13]. For the insertion of S4D and mAID sequences, we prepared *Satb1$^{+/Venus}$* zygotes by in vitro fertilization, using sperm from *Satb1$^{Venus/Venus}$* male mice and oocytes from C57BL/6NJcl mice. The following molecules were microinjected into the zygote: crRNA; tracrRNA; mRNA encoding Cas9; and ssODN encoding S4D (87 bp) or mAID (210 bp) with a hinge (GS) sequence flanked by 100 bp of homology region on both the 5′ and 3′ sides. Similarly, for the insertion of mAID sequence into N-terminal end of Bcl11b protein, an ssODN encoding the mAID was designed, followed by a hinge (GS) sequence flanked by 73 and 67 bp homology region on the 5′ and 3′ ends, respectively. In order to insert the mAID sequence into C-terminal end of the PD-1 protein, an ssODN composing of 40 bp 5′-homology region, hinge (LESGGGG) region, mAID and 37 bp 3′-homology region was designed. Sequences for guide RNA and ssODN used for these gene editing are listed in Supplementary Table 1.

### Administration of 5-Ph-IAA, 5-Ad-IAA, and pomalidomide into mice
5-Ph-IAA was synthesized as previously described[7] and purchased from Tokyo Chemical Industry. 5-Ph-IAA, 5-Ad-IAA, and other auxin derivatives were prepared in PBS at a concentration of 0.5 mg/mL. Aliquot of 200 μL of these auxin derivatives (0.1 mg) were administrated intraperitoneally into adult, pregnant, or lactating mice.

POM and 5-OH-POM were purchased from Tokyo Chemical Industry and Enamine, respectively. POM and 5-OH-POM were dissolved in dimethyl sulfoxide (DMSO, Sigma-Aldrich) to create a 50 mg/mL stock solution, which was stored at −20 °C. POM and 5-OH-POM were diluted at a final concentration of 1 mg/mL in PBS with 10% DMSO or in 15% DMSO, 17.5% Cremophor EL, 8.75% Ethanol, and 8.75% HCO-40. Subsequently, 200 μL of POM solution (0.2 mg) was intraperitoneally injected into adult, pregnant, or lactating mice.

### Flow cytometry
For flow cytometric analysis, cells were stained with the following fluorophore-conjugated antibodies: CD4 (RM4-5), CD8α (53-6.7), CD8β (YTS136.7.7), CD11b (M1/70), CD19 (1D3), CD21 (7G6), CD23 (B3B4), CD24 (M1/69), CD25 (PC61.5), CD44 (IM7), CD45 (30-F11), CD45.2 (104), CD45R (RA3-6B2), CD69 (H1.2F3), CD90.2 (53-2.1), CD103 (2E7), CD117 (2B8), CXCR4 (2B11), CXCR5 (2G8), IL-33R (ST2) (U29-93), KLRG1 (2F1), PD-1 (29F.1A12), TCRβ (H57-597), Bcl11b (25B6), Foxp3 (FJK-16s), Granzyme B (QA16A02), and IFN-γ (XMG1.2). Following surface antigen staining, the fixation and permeabilization buffers from the Transcription Factor Buffer Set (BD) were used to quench the intracellular EGFP fluorescent protein expressed from the *Rosa26* locus, following the manufacturer's protocol. The flow cytometry analyses were performed using a FACS CANTO II or a LSRFortessa (BD), and data were analyzed using FlowJo (BD) software. The gating strategies are shown in Supplementary Fig. 8.

### Cell preparation from tissues
Thymus, spleen, lymph nodes, peripheral blood, and Peyer's patches were harvested from mice and processed through 100 μm cell strainer to prepare single-cell suspensions. For the preparation of leukocytes in

the peripheral blood and spleen, red blood cells were lysed using ACK Lysing Buffer (Thermo Fisher Scientific).

For eliminating the majority of circulating T cells in the brain, the euthanized mouse was first perfused by injecting 20 mL of PBS into the left ventricle. The cerebrum and cerebellum were then collected and disaggregated through a 70 μm cell strainer, to obtain a single-cell suspension. Cells were centrifuged and resuspended in 4 mL of 37% Percoll (GE Healthcare) in PBS, which was carefully layered onto 4 mL of 70% Percoll, followed by 4 mL of 30% of Percoll, and finally topped with 2 mL of PBS, in a 15 mL conical tube. Cells were centrifugated at $350 \times g$ for 40 min, with brakes off, and the mononuclear cells, in between 37% and 70% Percoll, was harvested. To wash the cells, 3× volume of PBS was added and centrifuged at $350 \times g$ for 5 min. Cells were resuspended in PBS and subjected for flow cytometric analysis.

Lung cells were isolated as previously described[45]. Briefly, bronchoalveolar lavage fluid cells were rinsed with Hank's balanced salt solution containing 10% FBS using an 18-gauge plastic cannula attached to a 1 mL syringe. Subsequently, the lungs were excised and minced in Hank's balanced salt solution containing 10% FBS. The minced lungs were treated with 50 μg/mL of Liberase (Roche) and 1 μg/mL of DNase I (Roche) at 37 °C for 45 min. The digested lungs tissues were further dissociated using the gentleMACS (Miltenyi Biotec), and the isolated cells were collected by straining through a 40 μm cell strainer. Following red blood cell lysis with ACK Lysing Buffer (Thermo Fisher Scientific), immune cells were suspended in 30% Percoll PLUS (GE Healthcare) and centrifuged at $800 \times g$ for 30 min at 24 °C to remove epithelial cells. The purified immune cells were used for subsequent analysis after passing through a 37 μm filter.

Lamina propria lymphocytes were isolated from the small intestine. After removing feces and Peyer's patches, the small intestine was incubated in 20 mL of RPMI medium supplemented with 2% FBS and 5 mM EDTA at 200 rpm and 37 °C for 20 min. Following vigorous vortexing, the floating cells containing intraepithelial lymphocytes were discarded. The remaining tissues were pelleted, cut into small pieces, chopped with a razor blade, and incubated in 20 mL of the same digestion buffer containing 0.5 mg/mL of collagenase IV (Sigma-Aldrich) and 50 μg/mL of DNase I (Wako) at 200 rpm and 37 °C for 20 min. The digested cells were then pelleted and resuspended in 5 mL of 40% Percoll in PBS. This suspension was carefully layered onto 2 mL of 80% Percoll and centrifuged at $780 \times g$ for 20 min, with the brakes off. Mononuclear cells were harvested from the interface and washed for flow cytometry analysis.

For analyzing lung resident memory T cells, anaesthetized mice were injected intravenously with 1 μg biotin-conjugated anti-CD8β, 3 min before euthanization and tissue harvest. Lung tissues were minced and digested in collagenase D (Roche) for 30 min at 37 °C. Digested tissues were further dissociated in a 40 μM cell strainer. Following red blood cell lysis with ACK Lysing Buffer (Thermo Fisher Scientific), immune cells were suspended in 30% Percoll PLUS (GE Healthcare) and centrifuged at $800 \times g$ for 30 min at 25 °C to remove epithelial cells. The purified immune cells were used for subsequent analysis.

### CD4$^+$ T cell transplantation in *Rag1*-knockout mice

Spleens harvested from mice were mechanically dissociated using a 100 μm cell strainer to prepare single-cell suspensions. Red blood cells were then lysed using ACK Lysing Buffer (Thermo Fisher Scientific). CD4$^+$ T cells were selectively enriched using CD4 (L3T4) MicroBeads (Miltenyi Biotec), according to the manufactrurer's protocol. The purity of the isolated CD4$^+$ T cells was verified through flow cytometry, ensuring a composition of over 90% CD4$^+$ T cells. Subsequently, these cells were concentrated by centrifugation. Up to $5 \times 10^5$ cells were resuspended in 100 μL of PBS, and administered intravenously into *Rag1*-knockout mice.

### Protein degradation assays in cultured cells

The coding sequence of *OsTIR1$^{F74G}$* and *hCRBN* were cloned into the pMXs-IRES-hNGFR retroviral vector. The coding sequences of CXCR4-mAID and PD-1-mAID (mAID sequence fused to either the N-terminal or C-terminal end of PD-1) were cloned into the pMXs-IRES-eGFP and pMXs-IRES-DsRed retroviral vectors, respectively. The GP2-293 packaging cell line (Takara Bio, 631530) was utilized to produce viral-containing supernatants. Both HEK293T cells (Riken BRC, RCB2202) and Jurkat cells (Riken BRC, RCB0806) were resuspended in retrovirus-containing supernatant with 8 μg/mL polybrene and subjected to centrifugation at $2000 \times g$ for 90 min at 32 °C. POM (10 μM) and 5-Ph-IAA (1 μM) were added to the cell culture, and, where indicated on figures, cells were collected at specified time points for analysis by flow cytometry to detect degradation of target protein.

### Immunoblot

Total thymocytes were harvested from mice, and the cells were washed twice with ice-cold PBS. After pelleting the cells in a 1.5 mL tube, cells were lysed in lysis buffer (2% SDS, 50 μM Tris-HCl [pH 7.6] in PBS, supplemented with cOmplete Mini protease inhibitor cocktail [Roche]) at a concentration of $2 \times 10^7$ cells/mL, and incubated at 95 °C for 15 min. The samples were centrifugated at $13,000 \times g$ for 10 min and the supernatants were transferred to new 1.5 mL tubes. An equal volume of 2× Laemmli Sample Buffer (Bio-Rad) with β-mercaptoethanol was added to the lysates, followed by boiling at 95 °C for 5 min. For SDS–PAGE, lysates corresponding to $1 \times 10^5$ cells were loaded onto 10% polyacrylamide gels (ATTO) and the separated proteins were transferred onto polyvinylidene fluoride (PVDF) membranes (Bio-Rad). The membranes were blocked using TBST (Nacalai Tesque) containing 5% skim milk, then incubated with primary antibodies in blocking solution overnight at 4 °C. The membranes were incubated in HRP-conjugated secondary antibodies (Thermo Fisher Scientific) in blocking solution for 1 h at room temperature. Chemiluminescence was detected using an Amersham Imager 680 (GE Healthcare) after treatment with ECL Prime Western Blotting Detection Reagent (Cytiva). The primary antibodies used for immunoblotting include Satb1 (clone EPR3951, Abcam, ab109122), Ikaros (clone 4E9, Sigma-Aldrich, MABE912), Aiolos (clone 9D10, Sigma-Aldrich, MABE911), Actin (clone AC-40, Sigma-Aldrich, A4700), and Gapdh (clone 6C5, Santa Cruz Biotechnology, sc-32233).

### Proteomic analyses

Cell pellets were prepared for proteomics using the phase transfer surfactant (PTS) method[46,47]. Briefly, $1 \times 10^7$ cells were lysed with 400 μL PTS buffer containing 12 mM sodium deoxycholate (SDC), 12 mM sodium lauroylsarcosinate (SLS), and 100 mM Tris-HCl pH 9.0 and sonicated using a Cosmo BioRuptor on high power for five cycles of 1-min on, 1-min off. Protein concentrations were determined using a Pierce BCA Protein Assay Kit (Thermo Fisher Scientific). About 12 μg protein was digested for each sample. Bonded cysteine residues were reduced with 10 mM dithiothreitol (DTT) and then cysteines were alkylated with 40 mM iodoacetamide (IAA). Proteins were digested with 400 ng sequencing grade modified trypsin (Promega) at 37 °C, 750 rpm for 16 h. Afterwards, reactions were quenched and the concentration of SDC was decreased with the acidification of the samples with trifluoracetic acid to 0.5% concentration and addition then removal of ethyl acetate. The samples were dried using a centrifugal vacuum concentrator. Dried peptides were desalted with StageTips containing polystyrenedivinylbenzene (SDB-XC) resin[48].

For sample measurement, peptides were eluted from SDB-XC StageTips, vacuum dried, and then dissolved in 0.1% formic acid, 3% acetonitrile, 97% water. For each sample, 600 ng peptide was injected into the mass spectrometer. An Orbitrap Exploris 480 mass spectrometer coupled to an Ultimate 3000 liquid chromatography apparatus together with a Nanospray Flex ion source was used (Thermo Fisher Scientific). The internal diameter of the analytical column was 75 μm

and it was filled with 3 μm C18 particles to 12–15 cm length. Data was collected with data-independent acquisition (DIA) over 130 min. The solvents for liquid chromatography contained 0.1% formic acid and were of LC/MS grade, the gradient conditions were as follows with the percentage of acetonitrile as indicated: 0–2 min, 2.4%; 2–10 min, 2.4–8.0%; 10–100 min, 8.0–32.0%; 100–101 min, 32.0–72.0%; 101–110 min, 72.0%; 110–111 min, 72.0–2.4%; and 111–130 min, 2.4%. A positive ion spray voltage of 2.0 kV was applied, and the ion transfer tube was at 275 °C. The data acquisition settings were as follows[49]: a full mass spectrometry scan from 495 to 865 $m/z$; then 60 scans in DIA mode with isolation windows of 6 $m/z$ with overlapping window. All data collected was centroid, and a resolution of 30,000, 40% RF lens and a custom AGC target was applied throughout. For the DIA scans, HCD collision energies of 22, 26, and 30% were used.

The raw data files were processed using DIA-NN version 1.8 software[50] with MSFileReader software (Thermo Fisher Scientific) for file conversion. But first, A spectral library for the precursor ion identifications was made with DIA-NN using a *Mus musculus* (Taxonomy ID 10090) Swiss-Prot reviewed FASTA file database downloaded on 06/28/2022, and predicted FASTA digest with default settings, and with the additional option "--duplicate-proteins" so that entries in the sequence database with duplicate IDs were not skipped. Then the raw data files were searched using the spectral library with default parameters at 1% FDR, with the following additional options: --protein-qvalue 0.01, --peak-translation, --mass-acc-cal 10, --relaxed-prot-inf, --matrix-qvalue 0.01, --matrix-spec-q, --top 4. RT-dependent cross-run normalization was applied.

R package limma (version 3.58.1)[51] was used to determine the differential expression of proteins across each condition. Proteins that were undetectable in any replicates of a given condition were omitted from the analysis. The comparison was conducted between treatments (PBS vs 5-Ph-IAA and DMSO vs POM) within each degron system (OsTIR1-AID2 and hCRBN-S4D), as well as between Satb1 genotypes (*Satb1*[+/+] vs *Satb1*[V-AID/V-AID] and *Satb1*[+/+] vs *Satb1*[V-S4D/V-S4D]), in the context of each degron system (OsTIR1-AID2 and hCRBN-S4D) and treatment conditions (5-Ph-IAA and POM). The proteins that demonstrated an adjusted $p$ value threshold of <0.0005, were selected for further analysis.

### MC38 tumor model

The MC38 mouse colon adenocarcinoma cell line was originally provided by Dr. James P. Allison. Mice received intradermal injection of $5 \times 10^5$ MC38 cells into the right flank. Starting 10 days post-MC38 cell inoculation, mice were administered an intraperitoneal injection of either PBS or 200 μL of 0.5 mg/mL 5-Ph-IAA every 2 days. Tumor volumes were measured using calipers and accordingly to the following formula: tumor volume = π × (length × breadth × height)/6. On day 25, tumors were excised, weighed, and processed for flow cytometric analysis after digestion with 1.5 mg/mL collagenase (Wako) and 10 μg/mL DNase I (Roche) for 30 min on 37 °C.

In accordance with the animal experiment plan approved by RIKEN, euthanasia was promptly performed when excessive tumor growth (tumor diameter reaching 20 mm) was observed in this study. However, if the tumor shape was irregular and the long diameter exceeded 20 mm but the short diameter did not, euthanasia was also carried out based on assessments of changes in body weight or the impact of the tumor on the animal, such as difficulties in locomotion, feeding, or drinking.

### Pharmacokinetic analysis

Following intraperitoneal injection of 4 mg/kg 5-Ph-IAA and 5-Ad-IAA into 8-week-old C57BL/6 male mice, peripheral blood samples were collected at specified time points and plasma were prepared. The plasma was then mixed with nine times volume of acetonitrile to precipitate proteins and was centrifuged for 10 min at $19,000 \times g$. The supernatant was diluted with acetonitrile/0.1% formic acid (30/70) and applied for LC–MS/MS analysis. LC–MS/MS was performed using Vanquish UHPLC (Thermo Fisher Scientific) equipped with a COSMOCORE 2.6C18 column (2.1 mm × 50 mm, Nacalai Tesque) and TSQ-Vantage EMR triple quadrupole mass spectrometer (Thermo Fisher Scientific). Target compounds were separated by gradient elution at a flow rate of 0.4 mL/min [initially acetonitrile/0.1% formic acid (20/80) for 0.5 min then 20–98% acetonitrile for 3 min], and were detected in multiple reaction monitoring mode using the transitions of 5-Ph-IAA ($m/z$ 252 → 206) and 5-Ad-IAA ($m/z$ 310 → 135). TraceFinder software (Thermo Fisher Scientific) was used for peak area calculation.

### Statistics and reproducibility

Statistical analyses were conducted using GraphPad Prism 7 and R package stats (version 4.3.2). For comparing two independent groups, we employed a two-sample unpaired $t$-test. In cases involving multiple groups, we applied one-way analysis of variance (ANOVA) followed by multiple comparisons test using Tukey's "Honest Significant Difference" method, except for the proteomics analysis. For proteomics analysis, and where indicated in figure legends, one-way ANOVA followed by Dunnett's multiple comparisons test with a single pooled variance was performed on ligand-treated samples (5-Ph-IAA- or POM-treated) from *Satb1*[+/+], *Satb1*[V-S4D/V-AID], and *Satb1*[V-AID/V-AID] mice and were compared against their respective control samples (PBS- or DMSO-treated). Data are generally presented as mean ± standard error of the mean (s.e.m.). The sample sizes chosen were align with those commonly reported in literature.

### Reporting summary

Further information on research design is available in the Nature Portfolio Reporting Summary linked to this article.

### Data availability

The proteomics data are deposited to the ProteomeXchange Consortium via jPOST[52] partner repository with the dataset identifier JPST002981 (PXD050563 [https://proteomecentral.proteomexchange.org/cgi/GetDataset?ID=PXD050563] for ProteomeXchange). The *Rosa26*[OsTIR1/+], *Rosa26*[LSL-OsTIR1/+], *Rosa26*[OsTIR1-ΔEGFP/+], *Rosa26*[LSL-OsTIR1/+];*Pdcd1*[AID/AID]; *VavCre*, and *Rosa26*[LSL-OsTIR1/+];*Pdcd1*[AID/AID];*E8ICre* mouse strains are available at RIKEN BioResource Research Center (https://web.brc.riken.jp/en/) with the identifiers RBRC11884, RBRC11885, RBRC12401, RBRC12399, and RBRC12400, respectively. Source data are provided with this paper.

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

## Acknowledgements

We thank the members of Taniuchi Lab for helpful discussions. We thank Ms. Yuria Taniguchi, Ms. Chizuko Miyamoto, and Ms. Sawako

Muroi for plasmid construction, maintenance, and genotyping of mice. We thank the Support Unit for Bio-Material Analysis, RIKEN CBS Research Resources Division, and Dr. Masaya Usui for the measurement of compounds via LC–MS/MS analysis. We thank Dr. Shinya Hagihara and Dr. Shuhei Kusano at Molecular Bioregulation Research Team, RIKEN Center for Sustainable Resource Science for providing us with the auxin derivatives. We also thank Dr. Chisayo Kozuka and Dr. Azusa Inoue from Laboratory for Epigenome Inheritance, RIKEN Center for Integrative Medical Sciences for their assistance in observing mouse tissues using a fluorescent stereo microscope. This work was supported by Takeda Science Foundation to M.Y., JSPS KAKENHI (19H05747) to I.T., JSPS KAKENHI (JP21H04719 and JP23H04925), and JST CREST (JPMJCR21E6) to M.T.K., JSPS KAKENHI (23H04924) to K.I.

## Author contributions

M.Y. performed Satb1 study and wrote the manuscript. C.O. performed Bcl11b study. B.Z. performed PD-1 study under the supervision of S.F. T.K. performed Bcl11b study under the supervision of K.M. A.N. performed Satb1 study and helped with the preparation of the manuscript. C.B. and K.I. performed proteomics analysis under the supervision of J.S. C.Z. performed PD-1 study. S.Y. and T.S. contributed to the development of the hCRBN-S4D system. K.H. synthesized 5-Ph-IAA. Y.S. and F.S. performed pharmacokinetics analysis. M.T.K. contributed to the development of the OsTIR1-AID2 system. I.T. designed and supervised overall study, and wrote the manuscript.

## Competing interests

I.T., M.T.K., and K.H. are co-inventors to a patent application for *Rosa26*$^{OsTIR1(F74G)}$ transgenic mouse; patent applicant: RIKEN, National Institute of Genetics, and Okayama University of Science; name of inventors: I.T., M.T.K., and K.H.; application number: 2023-036211 (Examination, Japan); specific aspect of manuscript covered in patent application; Utility of *Rosa26*$^{OsTIR1(F74G)}$ transgenic mouse. The remaining authors declare no competing interests.
