## [Transparent Peer Review file · Nature Communications]

Cell-type specific, inducible and acute degradation of targeted protein in mice by two degron systems.

Corresponding Author: Dr Ichiro Taniuchi

Version 0:

Reviewer comments:

Reviewer #1

(Remarks to the Author)

Here, the authors employ two different degron systems to investigate the feasibility of acute targeted protein degradation in vivo in the mouse, in a cell type- and developmental stage-specific manner. Traditionally, targeted gene inactivation has been carried out through Cre-Lox systems, which operate at the chromatin level, commonly through excision of part of the coding sequence of the gene of interest (GOI). Though extremely useful, these systems have certain non-negligible drawbacks. Firstly, a considerable delay between induction of Cre-mediated gene deletion and depletion of the mRNA transcript of the GOI, which can continue to be translated, as well as of protein of interest (POI) itself. Secondly, the excision of the genomic sequence renders the process irreversible.

The comparatively recent development of methods targeting proteins have therefore presented an attractive alternative to Cre-Lox technologies. These methods are termed “degron systems,” as the POI is tagged with a peptide sequence, “degron tag.” This is recognised by a bivalent mediator molecule which concomitantly recognises a (subunit of) ubiquitin ligase, resulting in ubiquitination and rapid breakdown of the protein by the proteasome. Crucially, the bivalent molecule can be delivered at a timepoint of interest and its subsequent removal can result in reversal of POI depletion. Several of these methods are reviewed in Prozzillo et al. (2020).

Here, two different degron systems were tested: the auxin-inducible degron (AID) system, and the human cereblon (hCRBN) system. A drawback is that these systems do not employ the innate ubiquitination machinery of the cell to target POIs, and thus additional genetic modifications are required for the system to be functional. Specifically, the hCRBN degron system requires both tagging of the POI with the hCRBN-recognising SALL4 degron (S4D), as well as expression of hCRBN itself from the Rosa26 locus. The system is then dependent upon delivery of thalidomide or its derivatives (e.g. pomalidomide, POM) to induce ubiquitination and rapid degradation of the POI. The AID system requires introduction of the IAA17-derived degron tag, together with expression of plant TIR1 (OsTIR1) from the Rosa26 locus. Delivery of the plant hormone auxin (IAA) or one of its derivatives induces POI degradation. Here the authors describe the use of floxed alleles of the hCRBN and OsTIR1 expression cassettes to enable the use of Cre delete strains to allow tissue-specific activity of the degron systems. While this adds versatility to the systems, it necessitates the generation, maintenance, and crossing of multiple mouse lines to achieve the desired genotype. This is a time-consuming and costly approach. This can be largely avoided by the use of more direct degron tagging systems (reviewed by Hernández-Morán et al., 2024). Nonetheless, this paper represents a tour de force in mouse genetics that should be of interest to the readership of Nature Communications.

The current title of the manuscript (“Degron-based targeted protein degradation systems in mice”) is too general and does not reflect the specific technical achievements covered in the experimental results of the paper. The title should be amended to better reflect the content of the paper, by including more specific terms relating to the (blood-)cell-type specificity and variable developmental stage aspects of the experiments. We also recommend citing Hernández-Morán et al. (2024), which is the most up-to-date review on degron systems in vivo in mice and discusses the simpler, more straightforward systems available for use.

Due to the description and use of both systems, plus targeting of multiple POIs in different tissues and developmental stages, this has resulted in a very densely-written paper, which sometimes becomes difficult to follow. However, all of the information provided (including the negative data concerning the usage of alternative auxin derivatives) will constitute a valuable resource for researchers looking to utilise degron tags in vivo.

Specific comments:

- 1) What happens to the rest of the neo-substrates (other than Aiolos and Ikaros) when 5-OH-POM is used? Are their levels similarly less affected? Would it be possible to probe that with a western blot as done with Fig. 3d?
- 2) Page 9: since from the neonate experiments it is clear that POM and IAA readily cross the gut mucosa to target Satb1 within the neonates' thymic cells, did the authors consider using oral gavage instead of IP injections to target POI in adult mice as well?
- 3) Page 13: did the authors consider tagging the extracellular domain of the membrane proteins (perhaps just even in the cell lines)? Would this result in degradation of protein actively being trafficked to the membrane at the time of IAA/POM delivery?
- 4) Page 19, last paragraph of discussion: also include caveat that degraon-tagging the protein might in itself result in loss of protein functionality?
- 5) Fig. 4b: do the authors have any particular comments on the neo-substrates that were found to be shared between the two systems?
- 6) Page 7, lines 117-118: it is unclear to us how the (proposed) incomplete insolubility of POM in PBS/10% DMSO results in the sustained degradation of Satb1. Could the authors expand a bit, within a phrase or single sentence, on how this contributes? We would also appreciate a comment on the reproducibility of this sustained degradation by use of PBS/DMSO solvent. Do the authors therefore recommend using PBS/DMSO when someone using their system wants to induce sustained degradation of the POI over the span of multiple days/weeks?
- 7) Page 9, line 154: was recovery of protein levels in fetuses investigated at all?
- 8) Page 9, line 155: was dosage control a consideration when delivering IP injections to lactating mothers to target the P1 neonates? It is unlikely they will all be drinking the exact same amounts of milk.
- 9) Page 10, line 181: does this mean that the hCRBN system raises concerns for its usage generally?
- 10) Page 12, line 211-212: this tendency is statistically non-significant, yes?
- 11) Page 13, line 232: "efficacy" by which measure? Speed? % degradation?
- 12) Page 13, line 239: any suggestions as to why there was a "relatively spared PD-1+ cell fraction within the CD4+ TIL in 5-Ph-IAA-treated mice"?
- 13) Page 14, line 243: the usage of "enhanced" and "similar" to describe the two different antitumour responses described (expressing Cre in all blood cells vs. only CD8+ T cells) is contradictory. We suggest using only "similar" as the results in terms of tumour weight and size reduction are very close between the two.
- 14) Page 25, lines 263-265: unclear whether DN2, DN3, immature DP precursors and mature thymocytes are all part of the same differentiation "trajectory" – could clarify with a schema in figure 6?
- 15) Page 16, lines 279-282: considering the authors have previously mentioned that ILCs are not replenished, and that they are now suggesting (based on the recovery by postnatal week 9) that there is some sort of external replenishment of cells, is this finding dramatically contradictory to current dogma about how ILC progenitors function? Also, on line 282 the mention of "ILC progenitors" – are these the same or different from ILC2 progenitors (page 15, line 271)? It would be helpful to clarify or choose consistent nomenclature.
- 16) Page 16: would more injections during gestation (beyond 13.5 and 14.5) result in longer/more profound loss of lung ILC2s? (i.e. what is their developmental time window, are the authors capturing all of it with their current 2-day injection regime?)
- 17) Relevant to figures in main text and extended data: similar to what the authors did in Fig. 1c, it would be useful to have a dotted line at every histogram to indicate a cutoff between (Venus) positive and negative cells. Especially in the cases where the degradation efficacy is compared between different systems or timepoints, this would be very useful.
- 18) Extended figure 5a, b: how do these volcano plots differ from the ones depicted in Fig. 4a? It's not entirely clear.
- 19) Fig. 5d: are these data representative of >1 replicates?
- 20) Page 35, lines 722-725: should the authors correct two of the four brackets and replace "solvent cocktail" with "10% DMSO in PBS"?
- 21) Page 37, line 738: why such a difference between control and treatment groups (2 vs 7)? Also, is 3 and 3 enough on line 739?
- 22) Fig. 3a: are the panels under "mature thymocytes" a subset of the cells in the panels shown under "total thymocytes"? If so, could it be better indicated?
- 23) Page 9, section "Acute depletion of Satb1 mimics knockout" – please reference original KO paper in this section.

Reviewer #2

(Remarks to the Author)

In this study Dr. Taniuchi's group and collaborators developed two inducible and cell-type specific systems for targeted protein degradation in mice, using the auxin-inducible degron 2 and human cereblon (hCRBN)-SALL4 degron. The two systems are shown to be efficient for degradation of Satb1 in recapitulating the phenotype of Satb1 deficiency. The auxin system was also shown to be efficient for targeting PD1 as well as Bcl11b. These discoveries are important, given the use of new models of protein targeting. There are several issues which need to be addressed.

- it is impressive that after only 16hrs of administration of 5-Ph-IAA, they see a major reduction of the targeted protein, Satb1, with a max at 8hrs and recovery back at 72 hrs. Reduction in tissues (CNS) seems to be less efficient. How was the reduction in other tissues such as gut and lung?

- both the hCRBN-S4D and OsTIR1-AID2 systems reproduced the Foxp3 increase-related phenotype observed by Satb1 depletion with three administrations of ligands. These are very interesting observations. However do such fast changes in

Satb1, translate at physiological level? Any phenotype in terms of cellularity and disease signs?

- they claim that hCRBN-S4D and OsTIR1-AID2 systems target degradation of endogenous proteins. However that is not really the case, as the endogenous proteins need to be modified by introduction of degron target. This needs to be clarified.
 - it is not clear why they claim they identified new targets for hCRBN-S4D and OsTIR1-AID2 systems without having the degron target (mAID and S4D sequences) integrated at the specific loci, the same as they have for Satb1. This really needs to be clarified. If these are just targets correlated with Satb1 degradation, they should just be named accordingly, targets associated with Satb1 degradation, and NOT potential targets for hCRBN-S4D and OsTIR1-AID2 systems. How can they be neo-substrates in the hCRBN208 S4D degron system, when they do not have the mAID and S4D sequences integrated at those loci, but at Satb1? These conclusions really need to be re-evaluated and rephrased for accuracy.
 - also given the claims on the potential use in human treatment, there is need to discuss how this will work, as degron targets are not feasible for introduction in the human genome to achieve such goal.
 - they need to justify better the claim that the degron systems work better than the Cre system, as their degron system is also based on the Cre activity
 - nice results on tumor burden for PD1 targeting with systems controlled by Vavcre or E8ICre mice. CD8+ and CD4 (non-Treg) should be quantified separately (fig. 5f).
 - upon PD1 targeting, did the CD8+ T cells become more cytolytic?
 - the authors should review the study on the impact of Bcl11b in mature ILC2s: <https://pubmed.ncbi.nlm.nih.gov/26231117/>, published in 2015 in Immunity, and place their results with the Bcl11b degron in ILC2s in the context of that publication as well, particularly given that the study showed an impact of Bcl11b depletion in mature ILC2s in the lung, including on their function. Was there an impact of Bcl11b deletion on mature ILC2s, including on the function?
- Minor
- they should be more precise on the hCRBN and OsTIR1 constructs. In some places they describe as having GFP in others EGFP.

Reviewer #3

(Remarks to the Author)

In this manuscript, Yamashita et al explore the use of degron tagging technology in mouse models. Proof-of-concept for this approach has been demonstrated previously by several groups, but this manuscript builds on prior work in a number of important ways, including:

- By developing three novel mouse lines in which endogenous genes are tagged with degrons, they substantially increase the number of genes for which in vivo proof-of-concept for tag based degradation has been demonstrated in mice, showing broad applicability.
- By developing two new conditional alleles encoding E3 ligase receptors, they show cell-type restricted pharmacodynamic activity of degrader ligands following systemic administration in vivo. While this has been shown in other multicellular organisms previously, this is the first demonstration in mammals.
- They provide a method to control the kinetics of protein degradation and recovery in tissues by harnessing the depot effect; formulations with decreased ligand solubility take effect more slowly but persist for longer.
- They show robust evidence of tagged protein degradation in the brain using two independent approaches.
- They perform functional studies following acute degradation of Satb1, Bcl11b and Pcd1 in T cells

The results expand methods and knowledge on the in vivo use of degron tags and the conclusions are well supported by the experimental data. The authors should address the following points before publication.

- 1) The inclusion of an IRES:eGFP cassette downstream from the Tir1^{F74G} and hCRBN cDNAs provides a nice way to monitor expression of these constructs at the cellular level. Can the authors use this to show how broadly and heterogeneously these constructs are translated across cell types within tissues – e.g. in the brain, and in haematopoietic cells of different lineages? This would provide useful information for others wishing to use these lines.
- 2) It is inferred in the discussion (lines 331 – 333) that the F74G allele of OsTir1 alleviates problems with non-specific protein degradation without auxin treatment. But this is not quantified – can the authors quantify the in vivo expression of AID:Venus-tagged proteins in T cells that do versus do not express Tir1^{F74G}?
- 3) Related to point 2, it is not accurate to say that prior studies were unable to achieve systemic expression of the wildtype OsTir1 – this was achieved in reference 10 using the same LSL-based approach used in this paper.
- 4) Line 89: On insertion of AID or S4A tags - 'a slight decrease in Venus fluorescence intensity was observed'. This should be quantified as a % reduction in signal, as the extent of reduction is difficult to tell from Extended Data Figure 1c
- 5) Lines 90 – 93: The flow cytometry data in Extended Figure 1d make it appear that the ratio of CD4 to CD8 cells is affected by the AID tag in the non-induced state, but not the S4A tag. Please include measurements from the four independent experiments that were performed to determine if this effect is observed consistently.

Reviewer #4

(Remarks to the Author)

Version 1:

Reviewer comments:

Reviewer #1

(Remarks to the Author)

During the initial review of the manuscript, 23 specific comments and concerns were raised.

The authors have satisfactorily and comprehensively addressed all of the comments via the inclusion of the requested additional experiments and/or edits to the revised Manuscript.

Reviewer #2

(Remarks to the Author)

The authors addressed all the concerns of this reviewer. This is an interesting study and I am looking forward to seeing it published

Reviewer #3

(Remarks to the Author)

I thank the authors for their detailed responses and for adding data that significantly improve the quality of their manuscript. Congratulations on a very nice piece of work that advances the field of in vivo degron tagging. Andrew Wood

Reviewer #4

(Remarks to the Author)

made.

We thank the reviewers for taking their time to evaluate our manuscript and for recognizing the significance of our study. We have addressed most of the reviewers' concerns and corrected the format of the manuscript to comply with the Nature Communications submission guidelines.

Regarding the statistical analysis, we reported using an unpaired Student's t-test for comparisons between two groups; however, some of the analyses were incorrectly performed with unpaired Student's t-test with Welch's correction. We have revised the statistical analyses and now use an unpaired Student's t-test under the assumption of equal standard deviations. Importantly, this reanalysis did not change the conclusions of our study.

Additionally, we have noted that another paper using an *in vivo* degron system with OsTIR1^{F74G}-AID has been recently published (doi 10.1038/s41590-024-01933-7). While that study focuses on the biological consequences of acute depletion of transcription factors regulating early B cell development, our research is focusing on the characterization of two *in vivo* degron systems and includes its application to both cytoplasmic and membrane protein degradation in several tissues such as brain, thymus, lung of adult, neonatal and fetal mice. We have cited the above study in our revised manuscript (Ref 11), and believe it is complementary to our work, with regards to showing usefulness of degron system.

Reviewer #1 (Remarks to the Author):

Here, the authors employ two different degron systems to investigate the feasibility of acute targeted protein degradation *in vivo* in the mouse, in a cell type- and developmental stage-specific manner. Traditionally, targeted gene inactivation has been carried out through Cre-Lox systems, which operate at the chromatin level, commonly through excision of part of the coding sequence of the gene of interest (GOI). Though extremely useful, these systems have certain non-negligible drawbacks. Firstly, a considerable delay between induction of Cre-mediated gene deletion and depletion of the mRNA transcript of the GOI, which can continue to be translated, as well as of protein of interest (POI) itself. Secondly, the excision of the genomic sequence renders the process irreversible.

The comparatively recent development of methods targeting proteins have therefore presented an attractive alternative to Cre-Lox technologies. These methods are termed "degron systems," as the POI is tagged with a peptide sequence, "degron tag." This is recognised by a bivalent mediator molecule which concomitantly recognises a (subunit of) ubiquitin ligase, resulting in ubiquitination and rapid breakdown of the protein by the proteasome. Crucially, the bivalent molecule can be delivered at a timepoint of interest and its subsequent removal can result in reversal of POI depletion. Several of these methods are reviewed in Prozzillo et al. (2020).

Here, two different degron systems were tested: the auxin-inducible degron (AID) system, and the human cereblon (hCRBN) system. A drawback is that these systems do not employ the innate ubiquitination machinery of the cell to target POIs, and thus additional genetic modifications are required for the system to be functional. Specifically, the hCRBN degron system requires both tagging of the POI with the hCRBN-recognising SALL4 degron (S4D), as well as expression of

hCRBN itself from the Rosa26 locus. The system is then dependent upon delivery of thalidomide or its derivatives (e.g. pomalidomide, POM) to induce ubiquitination and rapid degradation of the POI. The AID system requires introduction of the IAA17-derived degron tag, together with expression of plant TIR1 (OsTIR1) from the Rosa26 locus. Delivery of the plant hormone auxin (IAA) or one of its derivatives induces POI degradation. Here the authors describe the use of floxed alleles of the hCRBN and OsTIR1 expression cassettes to enable the use of Cre delete strains to allow tissue-specific activity of the degron systems. While this adds versatility to the systems, it necessitates the generation, maintenance, and crossing of multiple mouse lines to achieve the desired genotype. This is a time-consuming and costly approach. This can be largely avoided by the use of more direct degron tagging systems (reviewed by Hernández-Morán et al., 2024). Nonetheless, this paper represents a tour de force in mouse genetics that should be of interest to the readership of Nature Communications.

We thank the reviewer for their very positive comments on our study.

The current title of the manuscript (“Degron-based targeted protein degradation systems in mice”) is too general and does not reflect the specific technical achievements covered in the experimental results of the paper. The title should be amended to better reflect the content of the paper, by including more specific terms relating to the (blood-)cell-type specificity and variable developmental stage aspects of the experiments. We also recommend citing Hernández-Morán et al. (2024), which is the most up-to-date review on degron systems in vivo in mice and discusses the simpler, more straightforward systems available for use.

We thank the reviewer for this important suggestion on the title of the manuscript on our study. We agree that our manuscript covers multiple proteins (Satb1, PD-1, Bcl11b, and CXCR4) targeted through different systems (OsTIR1-AID2 and hCRBN-S4D). The main message of this work is to showcase the establishment and comprehensive characterization of the *in vivo* degron systems, including their kinetics, cell- and developmental stage-specificity, and substrate specificity. Therefore, we have changed the title to “Cell-type specific, inducible and acute degradation of targeted protein in mice by two degron systems.” Additionally, as suggested, we have cited the review by Hernández-Morán et al. (2024) in the revised manuscript (Ref 12 in the revised manuscript).

Due to the description and use of both systems, plus targeting of multiple POIs in different tissues and developmental stages, this has resulted in a very densely-written paper, which sometimes becomes difficult to follow. However, all of the information provided (including the negative data concerning the usage of alternative auxin derivatives) will constitute a valuable resource for researchers looking to utilise degron tags in vivo.

We thank the reviewer again for their appreciation on our study.

Specific comments:

1) What happens to the rest of the neo-substrates (other than Aiolos and Ikaros) when 5-OH-POM is used? Are their levels similarly less affected? Would it be possible to probe that with a western blot as done with Fig. 3d?

In response to the reviewer's suggestion, we conducted immunoblot experiments to assess Zfp91 expression, following Satb1 depletion. As shown below, while there appears to be a trend towards Zfp91 degradation in the POM-treated condition compared to DMSO or 5-OH-POM, we are afraid that the quality of the blot is insufficient for publication. Therefore, we have decided to not include these results in the revised manuscript.

We also performed immunoblot experiments for Zmym2, as shown in the right panel. There was no detectable difference in the expression level of Zmym2 in the thymus of POM-treated *Rosa26^{hCRBN/+};Satb1^{V-S4D/V-S4D}* and *Rosa26^{hCRBN/+};Satb1^{+/+}* mice compared to DMSO-treated control mice. While Zmym2 was shown to be neo-substrates of hCRBN in cell line, our immunoblot addressed *in vivo* degradation of Zmym2 in the thymus. Thus, difference in experimental system would bring different results. It is possible that Zmym2 is degraded with high hCRBN expression and/or high POM concentration. It will, therefore, take more time and effort to clarify this point. But we believe that this is not an essential issue for our main conclusions in this study. We would appreciate the reviewer's understanding for not including additional immunoblots in the revised manuscript.

2) Page 9: since from the neonate experiments it is clear that POM and IAA readily cross the gut mucosa to target Satb1 within the neonates' thymic cells, did the authors consider using oral gavage instead of IP injections to target POI in adult mice as well?

We appreciate the reviewer's suggestion. In response, we conducted oral gavage experiments in adult mice, and the results have been added to Supplementary Fig. 3c of the revised manuscript. Oral gavage administration of 5-Ph-IAA resulted in a near-

complete depletion of *Satb1*^{V-AID} (3.5% of baseline levels) in *Rosa26*^{OsTIR1-ΔGFP/+};*Satb1*^{V-AID/V-AID} mice. On the other hand, oral gavage administration of POM in standard solvent only mildly reduced *Satb1*^{V-S4D} levels (46.2% of baseline) in *Rosa26*^{hCRBN/+};*Satb1*^{V-S4D/V-S4D} mice. This difference is likely due to differences in absorption and serum concentrations of the ligands. Importantly, intraperitoneal injection proved to be technically simpler and is likely to be more widely applicable. At any rate, information on the effects by oral gavage has been included in the results section of the revised manuscript as below:

“In addition to the intraperitoneal injection, oral gavage administration of POM and 5-Ph-IAA also induced *Satb1* degradation in peripheral blood T cells, though at varying degrees (**Supplementary Fig. 3c**). Orally administration of 5-Ph-IAA in *Rosa26*^{OsTIR1-ΔGFP/+};*Satb1*^{V-AID/V-AID} mice efficiently decreased the *Satb1*^{V-AID} levels, down to 3.5% of baseline, whereas orally administered POM in *Rosa26*^{hCRBN/+};*Satb1*^{V-S4D/V-S4D} mice showed less efficiency in reducing *Satb1*^{V-S4D} levels, falling down to only 46.2% of baseline.” (Page 9, Lines 150-155).

Please be noted that, for the oral gavage experiments, we used a newly established transgenic mouse line lacking ires-GFP in the *Rosa26*^{OsTIR1} allele, referred to as the *Rosa26*^{OsTIR1-ΔGFP} strain. The design and method for generating the *Rosa26*^{OsTIR1-ΔGFP} allele have been added to **Supplementary Fig. 1f** and are detailed in the Methods section (Page 25, Lines 565-569) of the revised manuscript. We believe that the *Rosa26*^{OsTIR1-ΔGFP} strain is useful for researchers who would like to use GFP expression for other usage in AID2 system.

3) Page 13: did the authors consider tagging the extracellular domain of the membrane proteins (perhaps just even in the cell lines)? Would this result in degradation of protein actively being trafficked to the membrane at the time of IAA/POM delivery?

We thank the reviewer’s for their thoughtful comment. We constructed a new retroviral vector encoding PD-1 with a mAID tag fused to its N-terminal end (extracellular domain) and investigated PD-1 degradation in transduced Jurkat cells, supplemented with 5-Ph-IAA in the culture medium. In this *in vitro* experiment, the addition of the mAID tag in the extracellular domain did not alter the cell surface levels of PD-1. We have included these findings in **Supplementary Fig. 6b** and in the results section of the revised manuscript as follows:

“Notably, the fusion of the mAID tag to the extracellular domain did not effectively degrade PD-1 (**Supplementary Fig. 6b**). These results highlight the importance of selecting the intracellular domain as the fusion site for the degraon tag when plasma membrane proteins are targeted for degradation.” (Page 15, Lines 252-255).

4) Page 19, last paragraph of discussion: also include caveat that degraon-tagging the protein might in itself result in loss of protein functionality?

We appreciate the reviewer's insightful suggestion. A recent study utilizing the *in vivo* OsTIR1^{F74G}-AID system (doi 10.1038/s41590-024-01933-7) reported phenotypic changes associated with the addition of a degron sequence to the endogenous genes. We now have added the following discussion to the revised manuscript:

"It should also be noted that the addition of a degron tag to the POI may impact its biological function, as recent studies have reported phenotypic changes caused by the degron-POI fusion (Ref 11). Therefore, optimizing the position of the degron tag in POI, as well as including linker sequences, should be carefully considered, and any resultant phenotypic changes should be thoroughly investigated." (Page 23, Lines 392-396)

5) Fig. 4b: do the authors have any particular comments on the neo-substrates that were found to be shared between the two systems?

We appreciate the reviewer's suggestion. We considered that proteins commonly decreased in the OsTIR1-AID2 and hCRBN-S4D systems were not neo-substrates. Rather, their decrease likely stems from Satb1 depletion. We believe that the reduction in Rag2 and Bcl6 levels are due to their decreased gene transcription following Satb1 depletion, as Satb1 positively regulates the transcription of both genes (Ref 36 and 37). This point has already been discussed in the original manuscript, as written below:

"Alongside Satb1, Rag2, and Bcl6 exhibited reduced levels in both degron systems. Given that Satb1 binds to the super-enhancer regions near the *Rag2* and *Bcl6* loci, positively regulating the expression of these genes (Ref 36 and 37), a decrease in Rag2 and Bcl6 protein levels is likely a consequence of a reduction in transcription through the attenuated activity of these enhancers upon loss of Satb1." (Page 19, Lines 323-326).

Additionally, we discussed other potential reasons for the reduction of another Satb1-dependent protein, Mta2, as follows:

"Mta2 also exhibited reduced levels in both degron systems. Based on the protein interaction between Satb1 and Mta2 (Ref 38), the degradation of Satb1 likely contributed to the instability of Mta2 proteins." (Page 19, Lines 326-328) in the revised manuscript.

6) Page 7, lines 117-118: it is unclear to us how the (proposed) incomplete insolubility of POM in PBS/10% DMSO results in the sustained degradation of Satb1. Could the authors expand a bit, within a phrase or single sentence, on how this contributes? We would also appreciate a comment on the reproducibility of this sustained degradation by use of PBS/DMSO solvent. Do the authors therefore recommend using PBS/DMSO when someone using their system wants to induce sustained degradation of the POI over the span of multiple days/weeks?

We thank the reviewer for bringing this point to our attention. We found that POM does not fully dissolve in the standard 10% DMSO/PBS solution and precipitates remain in the solvent. After intraperitoneal administration, POM aggregates were observed in the peritoneal cavity even after several days. However, a solvent cocktail composed of 15% DMSO, 17.5% Cremophor EL, 8.75% Ethanol, 8.75% HCO-40, and 50% PBS completely dissolved POM, with no precipitates or aggregation in the peritoneal cavity, following intraperitoneal administration. Therefore, when using the 10% DMSO/PBS solvent, the persistence of undissolved POM in the body is considered to be the cause of the sustained degradation. We clarified this in the revised manuscript:

“Undissolved POM formed aggregates in the peritoneal cavity following intraperitoneal injection, resulting in a prolonged POM supply that would cause sustained Satb1 degradation (**Supplementary Fig. 2a**).” (Page 8, Lines 125-127).

The reproducibility of sustained Satb1 degradation using the hCRBN-S4D system with 10% DMSO/PBS was confirmed across 4 biological replicates, all showing similar kinetics as shown in **Fig. 1f**, in the original manuscript.

As noted by the reviewer, the sustained degradation of POI with the hCRBN-S4D system using 10% DMSO/PBS could be advantageous for long-term depletion of POI. We have revised the discussion, as follows:

“Extended degradation of Satb1 over one week has been achieved using the hCRBN-S4D system, with the standard POM solution in 10% DMSO, PBS. Sustained depletion of POI could be maintained with weekly or bi-weekly administrations, which is beneficial for studying long-term POI depletion. Additionally, when using a solvent cocktail of 15% DMSO, 17.5% Cremophor EL, 8.75% Ethanol, 8.75% HCO-40, and 50% PBS for POM solution, the hCRBN-S4D system demonstrated sharper kinetics in POI depletion, compared to the OsTIR1-AID2 system.” (Page 20, Lines 334-339).

7) Page 9, line 154: was recovery of protein levels in fetuses investigated at all?

We appreciate the reviewer's comment. To address this, we used the OsTIR1/AID2 system to investigate the recovery of Satb1^{V-AID} in fetal DP thymocytes. To assess recovery in fetal cells, we performed an i.p. injection of 5-Ph-IAA (0.2 mL of 0.5 mg/mL) to pregnant mice at 15.5 dpc and measured Satb1 levels in fetal DP thymocytes at 18.5 dpc. Compared to the levels observed 1-day post-maternal i.p. administration (injection at 16.5 dpc and analysis at 17.5 dpc), Satb1 levels showed recovery within 3 days post-maternal i.p. administration. This recovery pattern is consistent with the kinetics observed following a single i.p. administration in adult mice. These results have been included in the revised manuscript:

"Similarly to adult mice receiving a single intraperitoneal administration of 5-Ph-IAA, Satb1^{V-AID} levels in *Rosa26^{OsTIR1/+};*Satb1^{V-AID/V-AID} fetuses also recovered within 3 days

post single intraperitoneal administration, in the pregnant mouse (**Supplementary Fig. 3g**). " (Page 11, Lines 171-174).

8) Page 9, line 155: was dosage control a consideration when delivering IP injections to lactating mothers to target the P1 neonates? It is unlikely they will all be drinking the exact same amounts of milk.

We thank the reviewer for their comment. In response to this question, we conducted additional experiments using different doses of 5-Ph-IAA administered to lactating mice and examined *Satb1*^{V-AID} levels in neonatal DP thymocytes. As expected, a lower dose (1/5x of the standard dose) resulted in less pronounced degradation of *Satb1*^{V-AID}. Contrastingly, a higher dose (5x of the standard dose) led to a greater degradation (2.7-3.9% of baseline), while, with the standard 0.1 mg dosage, we observed a slightly more variability in protein degradation (2.1-10.7% of baseline). This finding suggests that, at 0.1 mg administration, the effects of 5-ph-IAA, on POI degradation, can reach to a saturation point. Generally, levels below 10% of the baseline are considered to be sufficient for POI depletion, and we believe that the standard dose (0.2 mL of 0.5 mg/mL 5-Ph-IAA) is adequate for neonatal applications. We think these findings are informative. The results are presented in **Supplementary Fig. 3h** of the revised manuscript and described in the results section as follows:

"Varying the dose administration of 5-Ph-IAA in lactating mice resulted in differential levels of *Satb1*^{V-AID} in DP thymocytes of neonatal *Rosa26*^{OsTIR1/+}; *Satb1*^{V-AID/V-AID} mice (**Supplementary Fig. 3h**). Increasing the dose did not provide any significant benefit for *Satb1*^{V-AID} depletion in neonates, suggesting that the standard dosing (0.2 mL of 0.5 mg/mL 5-Ph-IAA) is sufficient for investigating POI depletion during the neonatal period." (Page 11, Lines 177-181).

9) Page 10, line 181: does this mean that the hCRBN system raises concerns for its usage generally?

We thank the reviewer for their valuable comment. As indicated by the proteomics analysis, the OsTIR1-AID2 system exhibits better substrate specificity, while the hCRBN-S4D system demonstrates sharper kinetics and a prolonged substrate degradation (latter as shown in **Fig. 1f**). Given that the neo-substrates of hCRBN, particularly the IKZF proteins, play a crucial role in the development and function of hematopoietic cells, the OsTIR1-AID2 system could be more beneficial for studying POI degradation in hematopoietic cells. However, in other cell lineages where the neo-substrates of hCRBN do not significantly impact on their differentiation or function, the hCRBN-S4D system might be an alternative choice, due to its advantage of sharp or prolonged kinetics.

These points were originally described in the initial manuscript, but we have now further clarified the advantages of prolonged degradation by the hCRBN-S4D system in the revised manuscript:

“Extended degradation of *Satb1* over one week has been achieved using the hCRBN-S4D system, with the standard POM solution in 10% DMSO, PBS. Sustained depletion of POI could be maintained with weekly or bi-weekly administrations, which is beneficial for studying long-term POI depletion. Additionally, when using a solvent cocktail of 15% DMSO, 17.5% Cremophor EL, 8.75% Ethanol, 8.75% HCO-40, and 50% PBS for POM solution, the hCRBN-S4D system demonstrated sharper kinetics in POI depletion, compared to the OsTIR1-AID2 system.” (Page 20, Lines 334-339).

10) Page 12, line 211-212: this tendency is statistically non-significant, yes?

We apologize for this confusing description in our initial manuscript. As the reviewer noted, there was no significant decrease in the levels of *Tmc8* and *Bbc3* in thymocytes of POM-treated *Rosa26^{hCRBN/+};Satb1^{V-S4D/V-S4D}* mice compared to DMSO-treated controls, using a cutoff of an adjusted p-value of <0.0005.

We could not conclude whether these proteins were neo-substrates of the OsTIR1-AID2 system or whether their reduction was due to secondary effects of *Satb1* depletion, as they also tended to decrease in the hCRBN-S4D system, albeit without reaching statistical significance. To clarify this point, we have added the adjusted p-values and fold changes of *Tmc8* and *Bbc3*, and the interpretation of the results to the main text as follows:

“Only *Tmc8* and *Bbc3* were specifically decreased in 5-Ph-IAA-treated *Rosa26^{OsTIR1/+};Satb1^{V-AID/V-AID}* mice and not in POM-treated *Rosa26^{hCRBN/+};Satb1^{V-S4D/V-S4D}* mice (Fig. 4c). These proteins also exhibited a tendency to decrease in the hCRBN-S4D system (*Tmc8*, logFC 0.23 with adjusted p-value 0.04; *Bbc3*, logFC 0.30 with adjusted p-value 0.32), raising the possibility of secondary effects of *Satb1* degradation.” (Page 14, Lines 235-238).

11) Page 13, line 232: “efficacy” by which measure? Speed? % degradation?

We appreciate the reviewer’s comment. In the original manuscript, we used the term “efficacy” to refer to the extent of degradation. In the cited study (doi: 10.1093/narcan/zcac019), cell surface expression of PD-1 was not completely lost after activation of the SMASh degron system, in both *in vitro* cell line experiments and in *ex vivo* cultured T cells from knock-in mice. However, in our study, PD-1 cell surface expression dropped to null levels, as observed from the *in vitro* cell line experiments (Fig. S6a) and on the T-follicular helper cells (Tfh) in 5-Ph-IAA-treated *Rosa26^{LSL-OsTIR1/+};Pdcd1^{AID/AID};VavCre* mice (Fig. S6d). Based on these findings, we concluded that the extent of degradation achieved by the OsTIR1-AID2 system is

significantly higher and more effective than that of the SMASh degron system. To improve clarity, we have rephrased the results section as follows:

“Notably, the extent of PD-1-mAID degradation appeared to be higher than that of a recently reported small molecule-assisted shutoff (SMASh) degron system (Ref 27). S” (Page 16, Lines 261-263).

12) Page 13, line 239: any suggestions as to why there was a “relatively spared PD-1+ cell fraction within the CD4+ TIL in 5-Ph-IAA-treated mice”?

We thank the reviewer for raising this crucial point. Currently, there is no evidence explaining why PD-1 degradation is less pronounced in Tregs, and this will be a focus to be addressed in future studies. However, we consider the importance in highlighting the potential differences in degradation levels among various cell types. We have added the following discussion to the revised manuscript:

“Although the degree of Satb1 degradation did not differ significantly among the T cell subsets examined, PD-1 degradation seemed to be less pronounced in Tregs than in conventional CD4⁺ T cells. Therefore, it is important to consider the POI degradation efficiency in each cell type.” (Page 23, Lines 396-398).

13) Page 14, line 243: the usage of “enhanced” and “similar” to describe the two different antitumour responses described (expressing Cre in all blood cells vs. only CD8⁺ T cells) is contradictory. We suggest using only “similar” as the results in terms of tumour weight and size reduction are very close between the two.

We thank the reviewer for this thoughtful suggestion. Accordingly, we have revised the description as follows:

“This selective PD-1 degradation in CD8⁺ T cells resulted in an anti-tumor response comparable to that observed in *Rosa26^{LSL-OsTIR1/+};Pdc1^{AID/AID};VavCre* mice in the MC38 xenograft model.” (Page 16, Lines 273-275).

14) Page 25, lines 263-265: unclear whether DN2, DN3, immature DP precursors and mature thymocytes are all part of the same differentiation “trajectory” – could clarify with a schema in figure 6?

We appreciate the reviewer's suggestion. For improved clarity, we have now included a schematic diagram of thymocyte differentiation to the bottom panel of **Fig. 6b** in the revised manuscript.

15) Page 16, lines 279-282: considering the authors have previously mentioned that ILCs are not replenished, and that they are now suggesting (based on the recovery by postnatal week 9) that

there is some sort of external replenishment of cells, is this finding dramatically contradictory to current dogma about how ILC progenitors function? Also, on line 282 the mention of “ILC progenitors” – are these the same or different from ILC2 progenitors (page 15, line 271)? It would be helpful to clarify or choose consistent nomenclature.

We thank the reviewer for this comment. In addition to the fetal liver, bone marrow also contains ILC2 progenitors. However, under physiological and steady-state conditions, lung ILC2s are predominantly generated from fetal liver ILC2 progenitors and are irreplaceable by those derived from bone marrow progenitors. It only when bone marrow ILC progenitors are transplanted into irradiated hosts, or when the host is challenged by helminth infection, that the bone marrow-derived ILC progenitors can effectively replenish lung ILC2 populations (doi 10.1126/science.aac9593, 10.1016/j.immuni.2020.09.002). In our experimental settings, the generation of fetal ILC2 progenitors was impaired by the temporal depletion of Bcl11b, leading to a reduced number of lung ILC2s at 3 weeks of age. However, we speculate that in the case when the lung niche for ILC2 is available or open, ILC progenitors from the bone marrow or tissue-resident progenitors can reconstitute ILC2s in the lung without irradiation or helminth infection, thus resulting in a comparable number of ILC2s by 9 weeks of age. Therefore, our observation aligns with the current understanding of ILC2 differentiation. We apologize for the confusing description in the original manuscript and have revised it as follows:

“These observations suggested that the reduced number of lung ILC2, which stems from the loss of Bcl11b expression during embryogenesis, leaves the lung niche open for bone marrow-derived or tissue-resident ILC progenitors to replenish ILC2s in adults.” (Page 19, Lines 312-315).

Regarding terminology, the "ILC progenitors" on Line 282 and "ILC2 progenitors" on Line 271 in the initial manuscript refer to the same population. In the revised manuscript, we have consistently used the term "ILC progenitors."

16) Page 16: would more injections during gestation (beyond 13.5 and 14.5) result in longer/more profound loss of lung ILC2s? (i.e. what is their developmental time window, are the authors capturing all of it with their current 2-day injection regime?)

We thank the reviewer for this suggestion. In response, we investigated whether extending the 5-Ph-IAA injecting for a period to 6 days, from 13.5 to 18.5 dpc, further decreases lung ILC2 number at 3 weeks of age. Interestingly, however, our findings showed no additional reduction in lung ILC2 number, in the group with Bcl11b depletion from 13.5 to 18.5 dpc. This suggests that Bcl11b-dependent ILC2 generation is time window-sensitive, with a critical developmental stage between 13.5 and 14.5 dpc, than from 15.5 dpc onwards. The results from this newly conducted experiment, as well as a comparison with the 13.5- 14.5 dpc depletion, are presented in **Supplementary Fig. 7b** and described in the main text as follows:

“Since fetal depletion of Bcl11b from 13.5 dpc to birth did not have an additive effect on reducing lung ILC2s (**Supplementary Fig. 7b**), we used 13.5 to 14.5 dpc depletion model for further analyses.” (Page 18, Lines 309-311).

17) Relevant to figures in main text and extended data: similar to what the authors did in Fig. 1c, it would be useful to have a dotted line at every histogram to indicate a cutoff between (Venus) positive and negative cells. Especially in the cases where the degradation efficacy is compared between different systems or timepoints, this would be very useful.

We thank the reviewer for this thoughtful suggestion. We have now added controls for no-venus histograms to **Supplementary Fig. 2d** and non-transduced histograms to **Supplementary Fig. 6a**, to better indicate the negative cutoffs. Additionally, our newly added figures (**Supplementary Fig. 1h**, **Supplementary Fig. 3b, c, g, h**, and **Supplementary Fig. 6b**) include no-venus or no-transduction control histograms. For **Fig. 6a**, Bcl11b-null population is not available for the analysis since all thymocytes express Bcl11b to some extent; therefore, no negative control histogram cannot be added.

18) Extended figure 5a, b: how do these volcano plots differ from the ones depicted in Fig. 4a? It's not entirely clear.

The volcano plot in **Supplementary Fig. 5a** displays the log fold-change on the x-axis and the negative log p-value on the y-axis. In contrast, **Fig. 4a** presents the log fold-change on the y-axis and average protein abundance on the x-axis, similar to an MA plot used for RNA-seq and microarray data. Both plots provide valuable information, and we would like to keep both **Fig. 4a** and **Supplementary Fig. 5a** in the revised manuscript.

19) Fig. 5d: are these data representative of >1 replicates?

The plots in **Fig. 5e** (**Fig. 5d** in the original manuscript) are representative of $n = 4$ each (PBS and 5-Ph-IAA) for *Rosa26^{LSL-OsTIR1/+};Pdc1^{AID/AID};VavCre*, and $n = 5$ each (PBS and 5-Ph-IAA) for *Rosa26^{LSL-OsTIR1/+};Pdc1^{AID/AID};E81Cre*. We apologize that the original figure legend lacked sufficient description of replicates for **Fig. 5e**. This has been now corrected in the revised manuscript.

20) Page 35, lines 722-725: should the authors correct two of the four brackets and replace “solvent cocktail” with “10% DMSO in PBS”?

We appreciate the reviewer for pointing this out. The description has been now corrected in the revised manuscript.

21) Page 37, line 738: why such a difference between control and treatment groups (2 vs 7)? Also, is 3 and 3 enough on line 739?

The discrepancy between the control and treatment groups occurred due to an imbalance in the genotypes following mouse crossing. In the revised manuscript, we have included additional biological replicates for the suggested conditions: hCRBN-S4D fetus data (n = 13 for the control group and n = 7 for the treatment group) and OsTIR1-AID2 neonatal data (n = 7 for the control group and n = 5 for the treatment group).

22) Fig. 3a: are the panels under “mature thymocytes” a subset of the cells in the panels shown under “total thymocytes”? If so, could it be better indicated?

As this reviewer pointed out, mature thymocytes are a subset of total thymocytes, defined by CD24^{lo}TCRβ⁺. In the revised manuscript, an arrow has been added to Fig. 3a to clearly indicate the population that refers to 'mature thymocytes'.

23) Page 9, section “Acute depletion of Satb1 mimics knockout” – please reference original KO paper in this section.

We thank the reviewer for the suggestion. We have now cited Kitagawa’s T cell-conditional Satb1 knockout mouse paper which reported Foxp3 depression in mature thymocytes and splenic T cells in the absence of Satb1 (Ref 14 in the revised manuscript). The description in the result section is amended as follows:

“Next, we investigated the *in vivo* effects of acute Satb1 depletion on immune phenotypes in mice, specifically focusing on the de-repression of *Foxp3*, which is a known Satb1-deficient phenotype observed in *Satb1^{Flox/Fdlox};Cd4Cre* mice (Ref 15: T cell-conditional Satb1 knockout study).” (Page 12, Lines 187-189).

Reviewer #2 (Remarks to the Author):

In this study Dr. Taniuchi’s group and collaborators developed two inducible and cell-type specific systems for targeted protein degradation in mice, using the auxin-inducible degron 2 and human cereblon (hCRBN)-SALL4 degron. The two systems are shown to be efficient for degradation of Satb1 in recapitulating the phenotype of Satb1 deficiency. The auxin system was also shown to be efficient for targeting PD1 as well as Bcl11b. These discoveries are important, given the use of new models of protein targeting. There are several issues which need to be addressed.

We thank the reviewer for their positive comments on our study.

- it is impressive that after only 16hrs of administration of 5-Ph-IAA, they see a major reduction of the targeted protein, Satb1, with a max at 8hrs and recovery back at 72 hrs. Reduction in tissues (CNS) seems to be less efficient. How was the reduction in other tissues such as gut and lung?

We thank the reviewer for the positive comments and for acknowledging the strength of the *in vivo* degron systems developed in this study. We also appreciate the valuable suggestions provided by this reviewer. In the revised manuscript, we have additionally examined Satb1 degradation in T cells residing in the lungs and the gut (small intestine lamina propria). As expected, Satb1 in both lung and gut T cells was effectively degraded 16 hours after intraperitoneal administration of 5-Ph-IAA. The results have been included in **Supplementary Fig. 3b** of the revised manuscript, and the main text has been amended as follows:

“After intraperitoneal ligand administration, successful Satb1^{Venus} degradation was observed in T cells within the spleen and lymph nodes, as well as in CD8⁺ T cells in the lungs and T cells in small intestine lamina propria (**Fig. 2b and Supplementary Fig. 3a, b**).” (Page 9, Lines 147-150).

For the experiments with the lungs and small intestines, we used a newly established transgenic mouse line lacking ires-EGFP in the *Rosa26*^{OstTIR1} allele, referred as the *Rosa26*^{OstTIR1-ΔEGFP} strain. The design and method for generating the *Rosa26*^{OstTIR1-ΔEGFP} allele have been added to **Supplementary Fig. 1f** and are detailed in the Methods section (Page 25, Lines 424-428) of the revised manuscript. We believe that the *Rosa26*^{OstTIR1-ΔEGFP} strain is useful for researchers who would like to use EGFP expression for other usage in AID2 system.

- both the hCRBN-S4D and OstTIR1-AID2 systems reproduced the Foxp3 increase-related phenotype observed by Satb1 depletion with three administrations of ligands. These are very interesting observations. However do such fast changes in Satb1, translate at physiological level? Any phenotype in terms of cellularity and disease signs?

We appreciate the reviewer’s comment on this key point and fully agree that understanding the physiological consequences of Satb1 degradation is crucial. However, in this manuscript, our focus is to comprehensively characterize the *in vivo* degron systems, including their kinetics, substrate specificities, and tissue and developmental stage-specificities, and validate the utility of the *in vivo* degron systems we developed. Therefore, we did not explore the physiological consequences of Satb1 depletion beyond Foxp3 de-repression in thymocytes and splenic T cells, and did not analyze the long term phenotypic changes, which requires more time and may require sequential Satb1 depletion for certain time windows, such as assessing the development of autoimmune diseases. In another unpublished study, we are addressing the effect of Satb1 loss, by conventional conditional knockout system for Satb1 deletion, in which the manuscript is under

revision. On the other hand, the use of acute POI depletion in inducing phenotypic changes has already been demonstrated in the PD-1 degron study (enhanced anti-tumor immunity) and the Bcl11b degron study (reduction of lung ILC2 in young mice) in the current manuscript. We hope that the reviewer understands that further exploration of disease development after Satb1 degradation is beyond the scope of this manuscript.

- they claim that hCRBN-S4D and OsTIR1-AID2 systems target degradation of endogenous proteins. However that is not really the case, as the endogenous proteins need to be modified by introduction of degron target. This needs to be clarified.

We thank the reviewer for highlighting this important point. As noted, the genes encoding POIs need to be modified to add the appropriate degron tag by knock-in technology. We used the term “endogenous” to emphasize that:

1. these genes are expressed under physiological transcriptional regulation, including normal transcription and splicing processes.
2. the addition of degron tags generally does not impact the function of the POI.
3. previous literatures have used similar strategies and employed the same terminology (Ref 8 [doi 10.1016/j.devcel.2022.03.013] and 10 [doi 10.7554/eLife.77987] in the manuscripts).

To avoid any potential confusion, we have refrained from using phrases such as “degradation of endogenous protein” in the revised manuscript (Page 3, Line 34; Page 5, Line 77; Page 6, Line 82).

- it is not clear why they claim they identified new targets for hCRBN-S4D and OsTIR1-AID2 systems without having the degron target (mAID and S4D sequences) integrated at the specific loci, the same as they have for Satb1. This really needs to be clarified. If these are just targets correlated with Satb1 degradation, they should just be named accordingly, targets associated with Satb1 degradation, and NOT potential targets for hCRBN-S4D and OsTIR1-AID2 systems. How can they be neo-substrates in the hCRBN208 S4D degron system, when they do not have the mAID and S4D sequences integrated at those loci, but at Satb1? These conclusions really need to be re-evaluated and rephrased for accuracy.

We thank the reviewer for their comment. Other proteins besides Satb1 may have been degraded due to their molecular similarities with the degron tags. For instance, the S4D tag is derived from the zinc finger of the SALL4 protein, and zinc finger proteins (such as IKZF proteins and ZFP91) are known to cross-react with hCRBN in a thalidomide derivative-dependent manner. Please be noted that IKZF proteins were also reduced in by POM injection, independently of Satb1 degradation, as shown in **Fig. 4d**.

Indeed, proteins that are commonly reduced in both the hCRBN-S4D and OsTIR1-AID2 systems are likely not neo-substrates of these degron systems, but instead are caused by the secondary effects of Satb1 degradation. The discussion regarding Rag2 and Bcl6, which have been addressed in the initial manuscript, describes potential effects of Satb1 depletion, on the transcriptional regulation of these genes as follows:

“Alongside Satb1, Rag2, and Bcl6 exhibited reduced levels in both degron systems. Given that Satb1 binds to the super-enhancer regions near the *Rag2* and *Bcl6* loci, positively regulating the expression of these genes (Ref 36 and 37), a decrease in Rag2 and Bcl6 protein levels is likely a consequence of a reduction in transcription through the attenuated activity of these enhancers upon loss of Satb1.” (Page 19, Lines 323-326).

Additionally, we discussed other potential reasons for the reduction of another Satb1-dependent protein, Mta2, as follows:

“Mta2 also exhibited reduced levels in both degron systems. Based on the protein interaction between Satb1 and Mta2 (Ref 38), the degradation of Satb1 likely contributed to the instability of Mta2 proteins.” (Page 19, Lines 326-328) in the revised manuscript.

For proteins which have been identified as specifically decreased in OsTIR1-AID2 system compared to hCRBN-S4D system, namely Tmc8 and Bbc3, we speculate these are false-positives and potentially secondary effects of Satb1 degradation, since they also tended to decrease in hCRBN-S4D systems, even though they did not reach any statistical significance with adjusted p-value cutoff of <0.0005. We amended the description and included interpretation of the results to improve clarity as follows:

“Only Tmc8 and Bbc3 were specifically decreased in 5-Ph-IAA-treated *Rosa26^{OsTIR1/+};[;]Satb1^{V-AID/V-AID}* mice and not in POM-treated *Rosa26^{hCRBN/+};[;]Satb1^{V-S4D/V-S4D}* mice (**Fig. 4c**). These proteins also exhibited a tendency to decrease in the hCRBN-S4D system (Tmc8, logFC 0.23 with adjusted p-value 0.04; Bbc3, logFC 0.30 with adjusted p-value 0.32), raising the possibility of secondary effects of Satb1 degradation.” (Page 14, Lines 235-238).

For the 13 proteins specifically decreased in the hCRBN-S4D system compared to the OsTIR1-AID2 system, it remains unclear which are true neo-substrates of the hCRBN-S4D system, and which resulted from secondary effects due to the depletion of Satb1 or other neo-substrates. Therefore, we have revised the following description to avoid overstatements, ensuring it does not imply that these proteins are definitively neo-substrates of the hCRBN-S4D system. The revised description is as follows:

“Thirteen proteins were decreased specifically in POM-treated *Rosa26^{hCRBN/+};**Satb1^{V-S4D/V-S4D}* mice, but not in 5-Ph-IAA-treated *Rosa26^{OsTIR1/+};**Satb1^{V-AID/V-AID}* mice.” (Page 14, Lines 231-232).

- also given the claims on the potential use in human treatment, there is need to discuss how this will work, as degron targets are not feasible for introduction in the human genome to achieve such goal.

We thank the reviewer for this valuable comment on the potential applications of this technology in the therapeutic development. Due to word count limitations, we did not include the following paragraph in the original manuscript, which discusses the possible use of *in vivo* degron technologies in the biomedical research. If space permits, we would like to add the following paragraph:

“In cases of adverse reactions occurring in systemic antibody administration, it is difficult to identify the cells responsible for the adverse effects, due to the systemic effects of the antibody. Immune checkpoint blockade therapy using anti-PD-1 or PD-L1 antibodies in cancer patients has been associated with autoimmune-like adverse effects, although the underlying mechanisms remain poorly understood (Ref 40 and 41). In the degron system, however, PD-1 degradation can be specifically induced in CD8⁺ T cells, regulatory T cells and monocyte-lineage cells by using *E81Cre*, *Foxp3Cre*, and *LysMCre*, respectively. Consequently, dissecting the effects of PD-1 degradation in each of these cell types, either individually or in combination, can potentially yield valuable insights into maximizing the effects of immune checkpoint blockade therapy while minimizing adverse effects. Additionally, raising a specific antibody against a G protein-coupled receptor, a process that is both cost- and time-consuming, is not always successful and is fraught with the problem of uncertain specificity. Given that genetic targeting of the degron module to the POI is a quick and reliable method for ensuring the specificity of proteins that lose function upon degradation, the degron system solves the above issues in antibody production, in particularly for the case of therapeutic applications.” (Page 22, Line 371-384).

- they need to justify better the claim that the degron systems work better than the Cre system, as their degron system is also based on the Cre activity

We thank the reviewer for the comment. Our intention was not to claim that the *in vivo* degron system is superior to gene knockout approaches including systemic or Cre-based conditional, as each technology has its own advantages and disadvantages. Compared to the Cre-loxP conditional gene inactivation approach, the

primary advantage of *in vivo* degron technology is its ability to achieve temporal and acute depletion of the POI, which is not easily achieved by conventional gene knockout strategies. This point has been further emphasized in the revised manuscript as follows:

“Furthermore, compared to irreversible genome engineering technologies including Cre-loxP-mediated conditional gene inactivation, one notable advantage of the degron system is its ability to recover protein expression within the same cells.” (Page 21, Lines 348-349).

- nice results on tumor burden for PD1 targeting with systems controlled by Vavcre or E8ICre mice. CD8⁺ and CD4 (non-Treg) should be quantified separately (fig. 5f).

We thank the reviewer for their positive comment and apologize for the confusions in figure labeling. In **Fig. 5d** (previously **Fig. 5f** in the initial manuscript), “Non-Treg” refers to Foxp3⁻ CD4 T cells (non-Treg CD4 T cells), which has now been labeled as “CD4 Tconv” for better clarity. Proportion of PD-1⁺ cells within CD8⁺ T cells in PBS and 5-Ph-IAA-treated conditions are newly included in **Fig. 5d** of the revised manuscript.

- upon PD1 targeting, did the CD8⁺ T cells become more cytolytic?

We thank the reviewer for raising this important point. Indeed, we observed that the Granzyme B⁺ and IFN-γ⁺ fractions were increased in PD-1-depleted CD8⁺ TILs, than PBS-treated CD8⁺ TILs, suggesting that PD-1-depleted CD8⁺ TILs became more cytolytic. These results have been included in **Fig. 5f** of the revised manuscript and are described in the results section as follows:

“Compared to PBS-treated controls, the frequency of Granzyme B⁺ and IFN-γ⁺ CD8⁺ effector TILs were increased in 5-Ph-IAA-treated mice. (**Fig. 5f**).” (Page 16, Lines 268-269).

- the authors should review the study on the impact of Bcl11b in mature ILC2s: <https://pubmed.ncbi.nlm.nih.gov/26231117/>, published in 2015 in Immunity, and place their results with the Bcl11b degron in ILC2s in the context of that publication as well, particularly given that the study showed an impact of Bcl11b depletion in mature ILC2s in the lung, including on their function. Was there an impact of Bcl11b deletion on mature ILC2s, including on the function?

We appreciate the reviewer for this comment. In response, we addressed whether acute depletion of Bcl11b from lung ILC2s affected their cell numbers and function by administering 5-Ph-IAA every other day for one week, into adult *Rosa26^{LSL-OsTIR1/LSL-OsTIR1};Bcl11b^{AID/AID};VavCre⁺ or -* mice. Analysis of lung ILC2 phenotypes showed no reduction in the number of lung ILC2 cells (as shown below) and no increase in the

Roryt⁺ in ST2⁺ ILC2 cells (as shown below). Although it is possible that a long-term depletion of Bcl11b may have a significant impact on ILC2 maturation and function, we believe that finding such conditions is beyond the scope of this manuscript.

Minor

- they should be more precise on the hCRBN and OsTIR1 constructs. In some places they describe as having GFP in others EGFP.

We apologize for the inconsistencies in the initial manuscript. We have now corrected these descriptions, and EGFP is consistently used throughout the revised manuscript.

Reviewer #3 (Remarks to the Author):

In this manuscript, Yamashita et al explore the use of degron tagging technology in mouse models. Proof-of-concept for this approach has been demonstrated previously by several groups, but this manuscript builds on prior work in a number of important ways, including:

- By developing three novel mouse lines in which endogenous genes are tagged with degrons, they substantially increase the number of genes for which in vivo proof-of-concept for tag based degradation has been demonstrated in mice, showing broad applicability.
- By developing two new conditional alleles encoding E3 ligase receptors, they show cell-type restricted pharmacodynamic activity of degrader ligands following systemic administration in vivo. While this has been shown in other multicellular organisms previously, this is the first

demonstration in mammals.

- They provide a method to control the kinetics of protein degradation and recovery in tissues by harnessing the depot effect; formulations with decreased ligand solubility take effect more slowly but persist for longer.

- They show robust evidence of tagged protein degradation in the brain using two independent approaches.

- They perform functional studies following acute degradation of *Satb1*, *Bcl11b* and *Pdcd1* in T cells

The results expand methods and knowledge on the in vivo use of degron tags and the conclusions are well supported by the experimental data. The authors should address the following points before publication.

We thank the reviewer for their positive comments on our study.

1) The inclusion of an IRES:eGFP cassette downstream from the *Tir1^{F74G}* and *hCRBN* cDNAs provides a nice way to monitor expression of these constructs at the cellular level. Can the authors use this to show how broadly and heterogeneously these constructs are translated across cell types within tissues – e.g. in the brain, and in haematopoietic cells of different lineages? This would provide useful information for others wishing to use these lines.

We appreciate the reviewer's valuable suggestion. In response, we confirmed *OsTIR1^{F74G}* transgene expression using a EGFP reporter in various tissues, including hematopoietic and lymphoid tissues, skin, brain, adipose tissue, heart, kidney, lung, gastrointestinal tract (stomach and small intestines), liver, and reproductive organs (uterus and ovaries). These results have been included in **Supplementary Fig. 1g** of the revised manuscript.

2) It is inferred in the discussion (lines 331 – 333) that the *F74G* allele of *OsTir1* alleviates problems with non-specific protein degradation without auxin treatment. But this is not quantified – can the authors quantify the in vivo expression of AID:Venus-tagged proteins in T cells that do versus do not express *Tir1^{F74G}*?

We thank the reviewer for pointing this out. As suggested by this reviewer, and to evaluate basal degradation by *OsTIR1^{F74G}*, *Satb1* expression was assessed in DP thymocytes either in the presence or absence of *OsTIR1^{F74G}*. In the presence of *OsTIR1^{F74G}*, the Venus MFI of Venus-mAID-*Satb1* thymocytes was 100.4%, than that of the absence of *OsTIR1^{F74G}*, suggesting that the basal degradation by *OsTIR1^{F74G}* is negligible (**Supplementary Fig. 1h**). We now added the following description to the results section of the revised manuscript:

“Consistent with previous reports, *OsTIR1*^{F74G} exhibited negligible basal degradation, as indicated by the comparable *Satb1*^{V-AID} levels in the presence or absence of *OsTIR1*^{F74G} (**Supplementary Fig. 1h**).” (Page 7, Lines 108-110).

3) Related to point 2, it is not accurate to say that prior studies were unable to achieve systemic expression of the wildtype *OsTir1* – this was achieved in reference 10 using the same LSL-based approach used in this paper.

We thank the reviewer for pointing this out. As suggested, after carefully reviewing Macdonald's paper and recognizing that another recently published study (doi: 10.1038/s41590-024-01933-7, Ref 11 in the revised manuscript) reported a mouse strain with systemic expression of *OsTIR1*^{F74G} from the *Rosa26* locus, we decided to omit the corresponding discussion from the revised manuscript.

4) Line 89: On insertion of AID or S4A tags - ‘a slight decrease in Venus fluorescence intensity was observed’. This should be quantified as a % reduction in signal, as the extent of reduction is difficult to tell from Extended Data Figure 1c

We thank the reviewer for this suggestion. In response, we have included the quantitative measurements of *Satb1*^{V-S4D} and *Satb1*^{V-AID} MFI in the revised manuscript as follows:

“However, a decrease in Venus fluorescence intensity was observed, with the mean fluorescence intensity (MFI) at 67.7% and 67.6% in CD4⁺CD8⁺ double-positive (DP) thymocytes of *Satb1*^{V-S4D/V-S4D} and *Satb1*^{V-AID/V-AID} mice, respectively, compared to *Satb1*^{Venus/Venus} controls.” (Page 6, Lines 91-93).

5) Lines 90 – 93: The flow cytometry data in Extended Figure 1d make it appear that the ratio of CD4 to CD8 cells is affected by the AID tag in the non-induced state, but not the S4A tag. Please include measurements from the four independent experiments that were performed to determine if this effect is observed consistently.

We appreciate the reviewer for bringing this point to our attention. Four biological replicates of *Satb1*^{+/+} and *Satb1*^{V-AID/V-AID} are shown below (upper two rows). To validate the phenotypes, we increased the number of *Satb1*^{+/+} controls and *Satb1*^{V-AID/V-AID} mice. Additional data from *Satb1*^{V-AID/V-AID} mice are shown in the bottom row. We concluded that *Satb1*^{V-AID/V-AID} has slightly lower CD4 single-positive (SP) fraction and higher CD8SP fraction in mature thymocyte population. The summary graph has been presented in **Supplementary Fig. 1e** in the revised manuscript and included in the main text as follows:

“In *Satb1*^{V-AID/V-AID} mice, there was a lower fraction of CD4 single-positive (SP) mature thymocytes and higher fraction of CD8 SP mature thymocytes (**Supplementary Fig. 1d, e**).” (Page 6, Lines 94-95).

Although the exact cause of the observed difference is unclear, it is not related to the *Satb1*-deficient phenotype and does not affect the conclusions of this study.

Nonetheless, we recognize that the insertion of a degron sequence into endogenous genes could potentially alter the function of the target protein, leading to phenotypic changes, as exemplified by a recent study utilizing the *in vivo* *OstTIR1^{F74G}*-AID system (doi 10.1038/s41590-024-01933-7). Therefore, we have now added the following discussion to the revised manuscript:

“It should also be noted that the addition of a degron tag to the POI may impact its biological function, as recent studies have reported phenotypic changes caused by the degron-POI fusion (Ref 11). Therefore, optimizing the position of the degron tag in POI, as well as including linker sequences, should be carefully considered, and any resultant phenotypic changes should be thoroughly investigated.” (Page 23, Lines 392-398)

Reviewer #4 (Remarks to the Author):

We appreciate the reviewer for taking their time to evaluate our manuscript and provide insightful suggestions for improvements.